# A molecular switch from STAT2-IRF9 to ISGF3 underlies interferon-induced gene transcription

Ekaterini Platanitis [1], Duygu Demiroz[1,5], Anja Schneller[1,5], Katrin Fischer[1], Christophe Capelle[1], Markus Hartl [1], Thomas Gossenreiter [1], Mathias Müller[2], Maria Novatchkova[3,4] & Thomas Decker [1]

Cells maintain the balance between homeostasis and inflammation by adapting and integrating the activity of intracellular signaling cascades, including the JAK-STAT pathway. Our understanding of how a tailored switch from homeostasis to a strong receptor-dependent response is coordinated remains limited. Here, we use an integrated transcriptomic and proteomic approach to analyze transcription-factor binding, gene expression and in vivo proximity-dependent labelling of proteins in living cells under homeostatic and interferon (IFN)-induced conditions. We show that interferons (IFN) switch murine macrophages from resting-state to induced gene expression by alternating subunits of transcription factor ISGF3. Whereas preformed STAT2-IRF9 complexes control basal expression of IFN-induced genes (ISG), both type I IFN and IFN-γ cause promoter binding of a complete ISGF3 complex containing STAT1, STAT2 and IRF9. In contrast to the dogmatic view of ISGF3 formation in the cytoplasm, our results suggest a model wherein the assembly of the ISGF3 complex occurs on DNA.

[1] Max Perutz Labs (MPL), University of Vienna, Vienna 1030, Austria. [2] Institute of Animal Breeding and Genetics, University of Veterinary Medicine Vienna, Vienna 1210, Austria. [3] Institute of Molecular Biotechnology of the Austrian Academy of Sciences (IMBA), Vienna 1030, Austria. [4] Research Institute of Molecular Pathology (IMP), Vienna Biocenter (VBC), Vienna 1030, Austria. [5]These authors contributed equally: Duygu Demiroz, Anja Schneller. Correspondence and requests for materials should be addressed to T.D. (email: thomas.decker@univie.ac.at)

Host defense by the innate immune system requires the establishment of antimicrobial states allowing cells to cope with microorganisms before the onset of the adaptive immune response. Interferons (IFN) are of vital importance in establishing cell-autonomous antimicrobial immunity. Particularly, the type I-IFN species IFN-α and IFN-β (collectively called IFN-I) or type III IFN (IFN-λ) are tightly associated with the antiviral state enabling cells to inhibit viral propagation. On the other hand, type II IFN (IFN-γ), while similarly capable of inducing the antiviral state, functions predominantly as a macrophage-activating cytokine[1,2].

The accepted scenario of signal transduction by the activated IFN-I receptor complex requires the Janus kinases TYK2 and JAK1 to phosphorylate the signal transducers and activators of transcription STAT1 and 2 on tyrosine[3,4]. SH2 domain-mediated heterodimerization enables STAT1–STAT2 to enter and reside in the cell nucleus. A third subunit, the interferon regulatory factor 9 (IRF9) joins the heterodimer to complete transcription factor ISGF3 which translocates to the nucleus and binds to interferon-stimulated response elements (ISREs) in ISG (interferon-stimulated gene) promoters[4,5]. The IFN-γ receptor on the other hand employs JAK1 and JAK2 to phosphorylate STAT1. STAT1 homodimers, a.k.a. gamma interferon-activated factor (GAF), translocate to the nucleus and stimulate ISG expression by binding to gamma interferon-activated sites (GAS)[6–8]. In addition to the canonical ISGF3 and GAF, experiments in knockout cells suggest that transcription factors containing IRF9 and either STAT1 or STAT2, but not both, have the potential to control ISG expression[9–11]. Furthermore, transcriptional activity of an ISGF3 complex assembled from unphosphorylated STATs (uSTATs) has been proposed[12]. The extent to which such noncanonical complexes form and control ISG expression under conditions of a wild-type cell remains elusive and is an important aspect of this study.

As the emergence of cell-autonomous immunity is an arms race between pathogen replication and restrictive mechanisms of the host, speed is a crucial attribute of the cellular response to IFN. This is particularly true for antimicrobial gene expression that must rapidly switch between resting-state and active-state transcription. Mechanisms to meet this demand include remodeling and modification of promoter chromatin prior to the IFN response[13]. Moreover, a host of studies support the concept that cells permanently produce a small amount of IFN-I that stimulates a low, tonic signal by the IFN-I receptor[14,15]. This was shown to generate a baseline transcriptional response of IFN-induced genes (ISG[16]). The enhancement of ISG transcription by an IFN stimulus from basal to induced levels has been compared with the revving up of a running engine[17]. The "revving-up" or "autocrine loop" model predicts a tight coupling between homeostatic and receptor-mediated interferon signaling and that transcriptional ISG activation in resting and activated states differs in its intensity, but abides by the same mechanism.

Our study challenges this notion. By combining ChIP-seq and transcriptome analysis, we find that basal expression of many ISGs is controlled by a preformed STAT2–IRF9 complex, whose formation does not require signaling by the IFN-I receptor. IFN treatment induces a rapid switch from the STAT2–IRF9 to the canonical ISGF3 complex, revving up ISG transcription. Quantitative proteomic analysis and in vitro interaction studies suggest a model, wherein part of IRF9 resides in the nucleus under homeostatic conditions and the assembly of the ISGF3 complex occurs on DNA. In conclusion, combining high-throughput data enabled us to reveal mechanisms by which different states of the promoter-associated transcription factor ISGF3 control the switch from homeostatic to interferon-induced gene expression.

## Results

**Transcription-factor binding to ISG promoters**. We used macrophages to study the mechanisms contributing to constitutive ISG expression.

Basal expression of ISGs controlled via ISRE promoter sequences (Irf7, Usp18, and Oas1a) in bone marrow-derived macrophages (BMDM) was strongly reduced by ablation of any of the three ISGF3 subunits (Fig. 1a). Surprisingly, genes regulated predominantly by STAT1 dimer binding to GAS sequences, such as Irf1 and Irf8, were largely unaffected by the gene deficiencies including STAT1 (Fig. 1b). Since IRF9 is the DNA-binding subunit of all ISRE-associated transcription factors, our results point toward an important role for this protein and its associates for basal ISG expression. Consistent with this, RNA-seq and gene set enrichment analysis (GSEA; Fig. 1c) underscore the impact of IRF9 loss on global ISG transcription in resting BMDM. To determine whether IRF9 dependence reflected the formation of an ISGF3 complex, we performed ChIP-seq in wt BMDM. The integrated experimental approach is shown in Supplementary Fig. 1. This examination of all three ISGF3 subunits simultaneously in murine cells was made possible by the generation of an anti-IRF9 monoclonal antibody, 6FI-H5, which yields excellent signal-to-noise ratios in ChIP (Supplementary Fig. 2a). IRF9 dependence of STAT1/2 binding was confirmed by ChIP-seq in Irf9$^{-/-}$ BMDM.

The combination of RNA-seq (plotting wt vs. Irf9$^{-/-}$ cells) and ChIP-seq data (revealing promoter occupancy in wt BMDM) in a scatterplot showed promoters of a majority of IRF9-dependent genes associated with STAT2 and IRF9, but not with STAT1 (Fig. 2a, d). A quantitative representation of promoters binding STAT2–IRF9, ISGF3, or STAT1 dimers is given in the pie chart inserts and genes are listed in Supplementary Data 1. A much smaller fraction was associated with all subunits of an ISGF3 complex. Alignment with tracks from a recently published ATAC-seq data set from resting BMDM[18] correlated STAT2–IRF9 peaks in a majority of promoters with open chromatin (Fig. 2 and Supplementary Fig. 2b). A brief treatment with either IFN-I or IFN-γ caused a vast majority of promoters, including many of those associated with STAT2–IRF9 in resting state, to bind ISGF3 (Fig. 2b, c and e, f). Promoters associated with STAT1 dimers were prominently represented among IFN-γ-induced genes, but not among IFN-I-induced ISG. According to the prevailing JAK-STAT paradigm, transcriptional IFN-I responses are ISGF3/ISRE based, while those to IFN-γ use STAT1 dimers/GAS. Surprisingly, we observed a prominent contribution of ISGF3 to IFN-γ-induced ISGs, as well as the de novo formation of STAT2–IRF9 complexes at both IFN-I- and IFN-γ-induced ISGs. Among the overlapping peak sets, a small number of STAT1–IRF9 complexes can be found, which, upon visual inspection, appeared to be co-bound by STAT2 and are therefore more likely the result of type II peak prediction error (Supplementary Fig. 2c). To examine whether the peak coincidence in ChIP-seq reflected simultaneous binding of the ISGF3 subunits to the same promoters, we used a ChIP-re-ChIP approach. In IFN-β-treated BMDM, both STAT1 and STAT2 were re-precipitated from a primary IRF9 ChIP. Reprecipitation of STAT1 and STAT2 from IRF9 ChIP in resting cells was below the detection limit for weak constitutive binders, such as the Irf7 promoter (Fig. 2a). In case of the Usp18 and particularly Oas1a promoters that show stronger constitutive binding, low amounts of STAT2, but not of STAT1, were re-precipitated (Supplementary Fig. 3a). ChIP-seq peak areas for promoters that were pre-associated with STAT2–IRF9, and in their majority, switched to ISGF3 after IFN-β treatment, contained the expected ISRE consensus (RRTTTCNNTTTYY[19]; Supplementary Fig. 3b). Comparing ISRE sequences between promoters associated with ISGF3

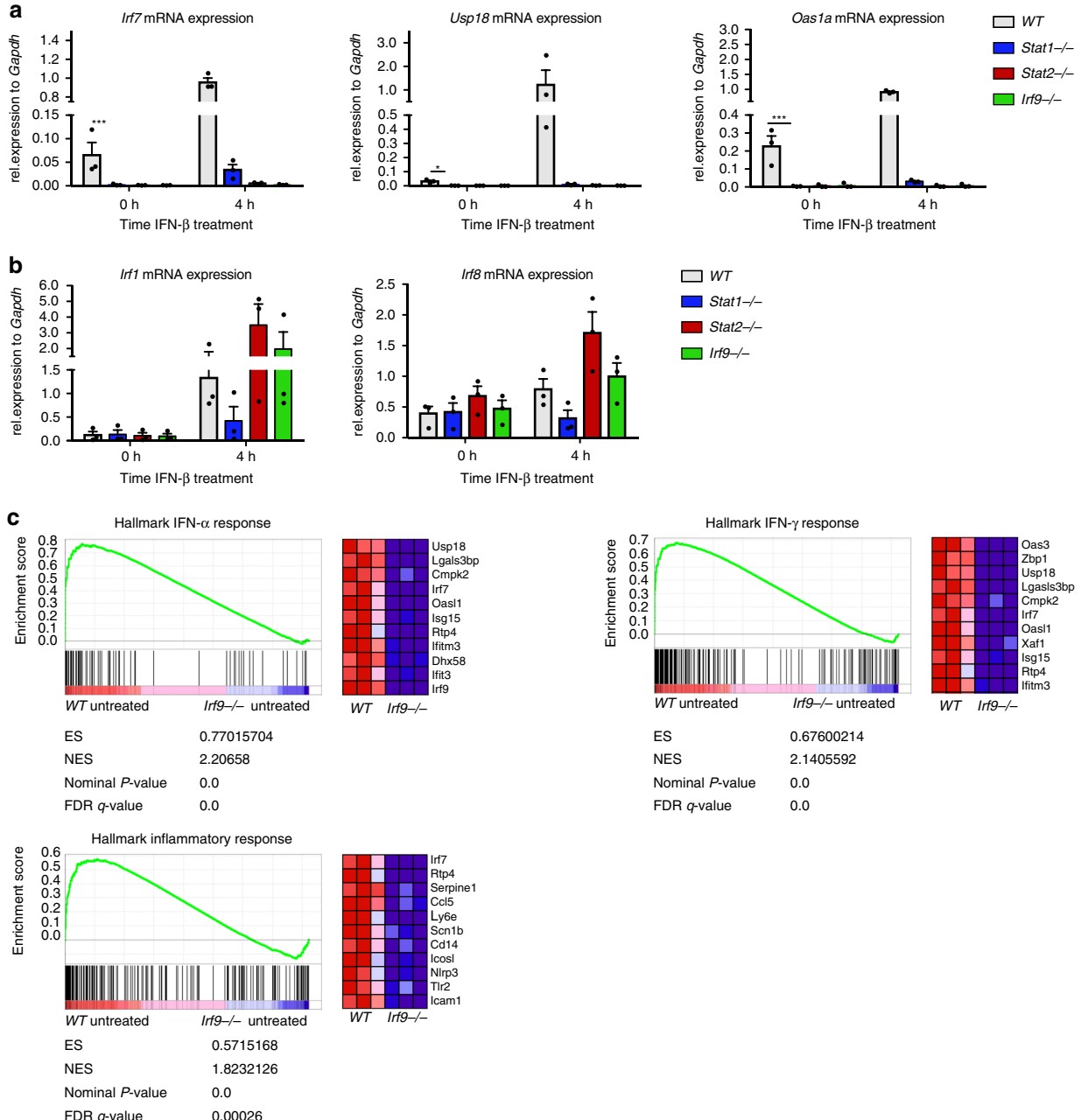

**Fig. 1** Conditions of basal ISG expression. **a**, **b** Bone marrow-derived macrophages (BMDM) isolated from wild-type (WT), $Stat1^{-/-}$, $Stat2^{-/-}$, and $Irf9^{-/-}$ mice were treated with 250 IU/ml of IFN-β as indicated. Gapdh-normalized gene expression was measured by RT-q-PCR. Data represent the mean and standard error of the mean (SEM) values of three independent experiments. P-values were calculated using the paired ratio t-test (*$P \leq 0.05$; **$P \leq 0.01$; ***$P \leq 0.001$). **c** Gene set enrichment analyses showing upregulation of an IFN and inflammatory response signature of untreated WT compared with untreated $Irf9^{-/-}$ BMDM. The top correlated genes for each biological triplicate are displayed in the corresponding heat maps. The total height of the curve indicates the extent of enrichment (ES), with the normalized enrichment score (NES), the false discovery rate (FDR), and the P-value. Source data are provided as a source data file

and STAT2–IRF9, respectively, after IFN-β treatment showed very similar binding preferences of both complexes with a somewhat larger preference of STAT2–IRF9 for ISREs containing a T nucleotide in position 14 (Supplementary Fig. 3c).

Taken together, the data are consistent with the notion that the switch from resting- to activated-state transcription is caused by a transition from STAT2–IRF9 to ISGF3 at a majority of ISG promoters. This finding was corroborated in mouse embryonic fibroblasts, where basal ISG expression was reduced by IRF9 deficiency in a gene set overlapping with IRF9-dependent genes in BMDM, both resting and type I-IFN treated (blue symbols in

Fig. 3a–d; Supplementary Data 2 and 3). Binding of STAT2 and IRF9 to ISG promoters was readily detectable in resting cells (Fig. 3e). Similar to BMDM, STAT1 binding was observed at a majority of ISGs only after treatment with IFN-β. To extend our analysis to human cells, THP-1 monocytes were rendered IRF9 deficient using CRISPR–Cas9 technology (Supplementary Fig. 4a, b). Similar to murine cells, the expression of many constitutive and IFN-β-induced genes was reduced by the Irf9 knockout (Supplementary Fig. 4c, d, Supplementary Data 4 and 5). The gene set showed some overlap with IRF9-dependent genes in BMDM, but many genes differed. We were not able to

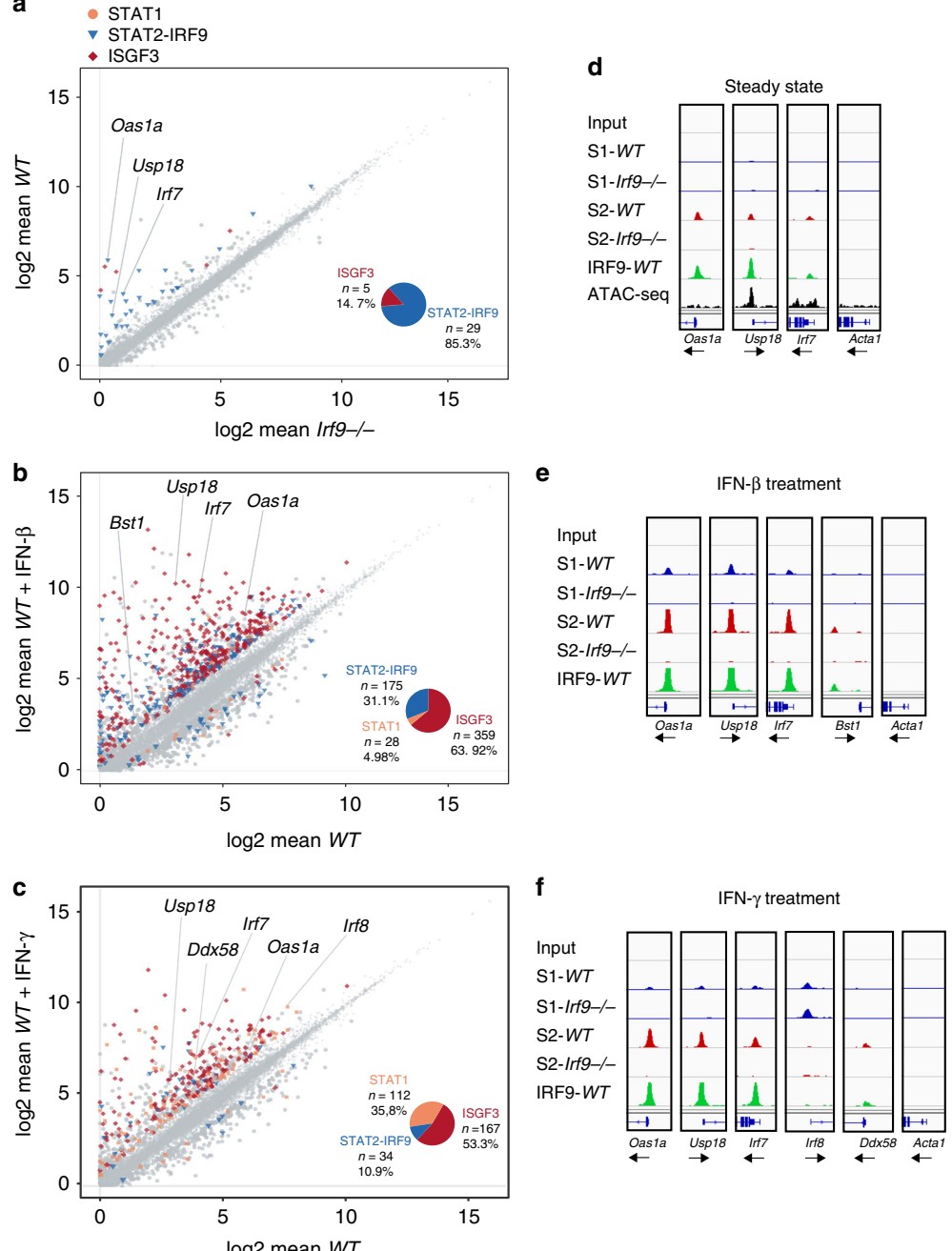

**Fig. 2** IFN-induced gene expression and STAT complexes in BMDM. **a–c** panels on the left. Scatterplot linking RNA-seq (n = 3) with ChIP-seq (n = 2) data (BMDM). Differentially expressed genes (log-fold change (lfc) > 1, padj < 0.05) between *Irf9*[−/−] and WT untreated (**a**), WT untreated versus WT IFN-β (**b**), or WT untreated versus IFN-γ-treated (**c**) BMDM are shown. Genes associated with complexes containing at least one of the ISGF3 subunits (STAT1, STAT2, and IRF9) according to ChIP with the respective antibodies are color-coded as follows. Blue triangles: STAT2–IRF9; red diamonds: STAT1, STAT2, and IRF9 (ISGF3); beige squares: STAT1 only. The pie chart inserts show the relative proportions of genes associated with STAT2–IRF9, ISGF3, or STAT1 dimers. Panels on the right **d–f**. Representative browser tracks of the ChIP-seq experiments shown in **a–c**). Data from untreated (**d**), IFN-β (90 min; **e**), or IFN-γ- (90 min; **f**) treated BMDM derived from wild-type (WT) and *Irf9*[−/−] (IRF9[−/−]) mice are shown (scale 0–150). Individual tracks represent binding of STAT1 (S1/blue), STAT2 (S2/red), and IRF9 (green). Tracks are shown for ChIP-seq and control input, as well as regulatory chromatin sites from ATAC-seq for untreated BMDM, derived from data in ref. [18]

convincingly demonstrate constitutive binding of any of the ISGF3 subunits (Supplementary Fig. 4e). Possible causes for this result are considered in the "Discussion" section.

**Spatial proximity and complex formation of ISGF3 subunits.** Co-immunoprecipitation studies in human epithelial cell lysates

concluded that STAT2 and IRF9 form complexes devoid of STAT1[20]. Corroborating this result, recent work solved the structure of the binding interface and determined that STAT2 binds IRF9 with 500-fold higher affinity than STAT1[21]. STAT2–IRF9 as well as STAT1–STAT2 complexes could be co-immunoprecipitated from resting-cell extracts[20,22–24]. Our results in mouse BMDM are in agreement with these observations.

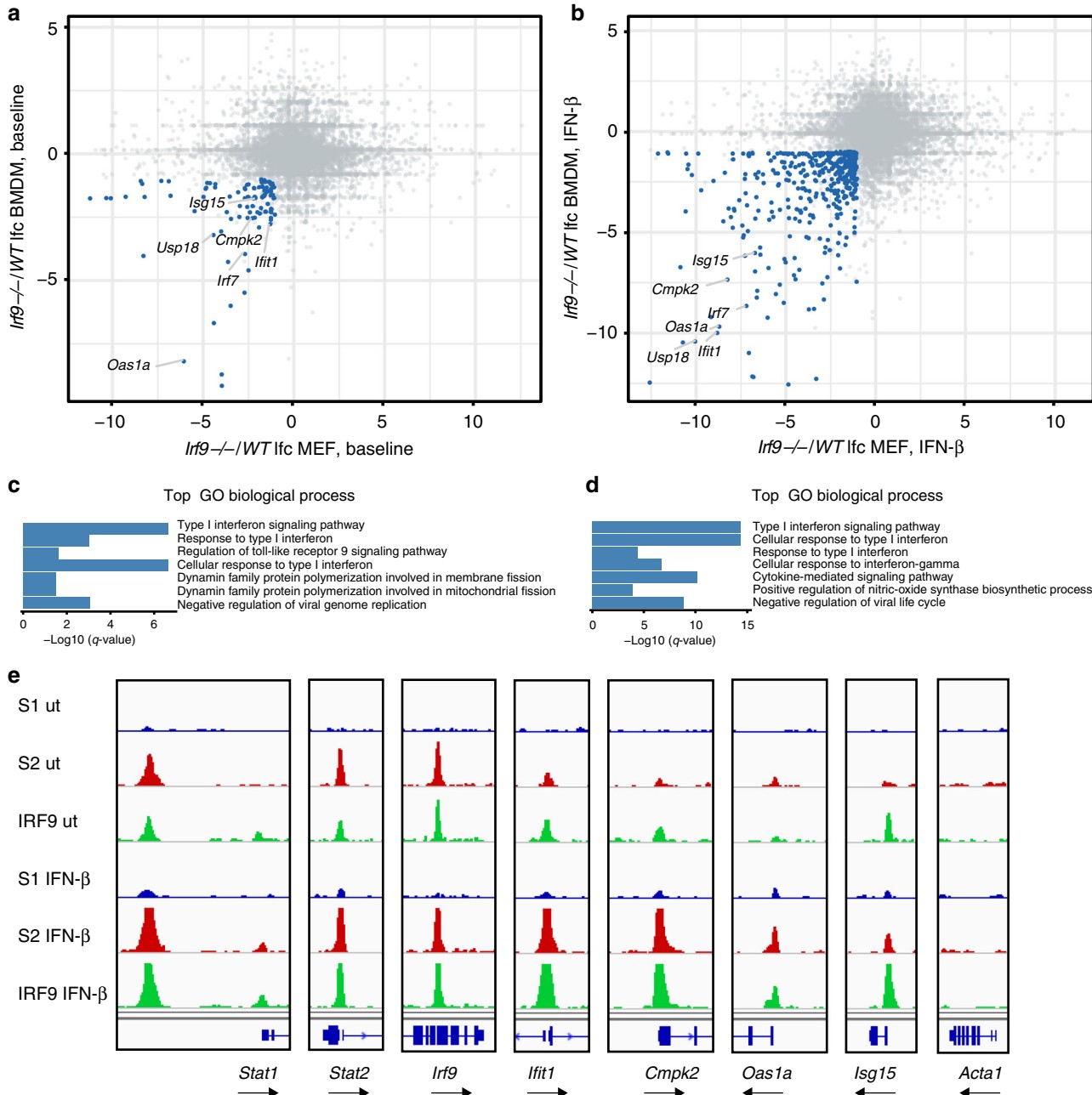

**Fig. 3** IFN-induced gene expression and STAT complexes in MEFs. **a**, **b** Log-fold change (lfc)/lfc plot comparing WT or IRF9$^{-/-}$ BMDM with mouse embryonic fibroblast (MEF). mRNA expression ($n = 3$) ratios, with a cutoff padj $\leq 0.05$ and lfc $\geq 1$, (IRF9$^{-/-}$/WT) in resting (**a**) or in IFN-β- treated (**b**) cells is plotted. Genes affected by the loss of IRF9 in both cell types are displayed in blue. **c**, **d** GO analysis of the IRF9-dependent genes in untreated (**c**) and IFN-β-treated (**d**) cells as defined in **a**, **b**. The top seven pathways are listed for IRF9-dependent genes (in blue). **e** Representative genome browser tracks for transcription factor binding at ISG and control loci (scale 0–60): STAT1 (S1/blue), STAT2 (S2/red), and IRF9 (green). Tracks represent ChIP-seq experiments in untreated and IFN-β-treated MEF

Reciprocal immunoprecipitations demonstrated a clear association between STAT2 and IRF9 in resting and IFN-I-treated BMDM (Fig. 4a, b). Furthermore, we observed a weak association between STAT2 and STAT1 that, in line with earlier observations, did not increase after IFN-I treatment[22,23]. STAT1 and IRF9 could not be coprecipitated, despite earlier studies assigning transcriptional activity to STAT1–IRF9 complexes[25,26]. Thus, while STAT2–IRF9 and a small amount of STAT1–STAT2 complexes just above the detection limit can be demonstrated in resting cells, we observed neither ISGF3 nor other complexes containing both STAT1 and IRF9. To corroborate these findings and to rule out that the data

reflected stability under IP conditions rather than complex formation in cells, we used the BioID proximity labeling technology as depicted in Supplementary Fig. 1. Modified biotin ligase BirA* fusion proteins[27] biotinylate proteins within a distance of ~10 nm[28]. Raw 264.7 macrophages were engineered to express doxycycline (Dox)-inducible, myc-tagged BirA* fusion genes with Stat1, Stat2, or Irf9. N-terminal fusion proteins were selected based on their ability to restore ISGF3 function in the respective knockout MEFs (Supplementary Fig. 5b, d, f). To avoid overexpression artifacts, we adjusted the Dox concentration to induce levels of the BirA* fusion proteins that closely matched

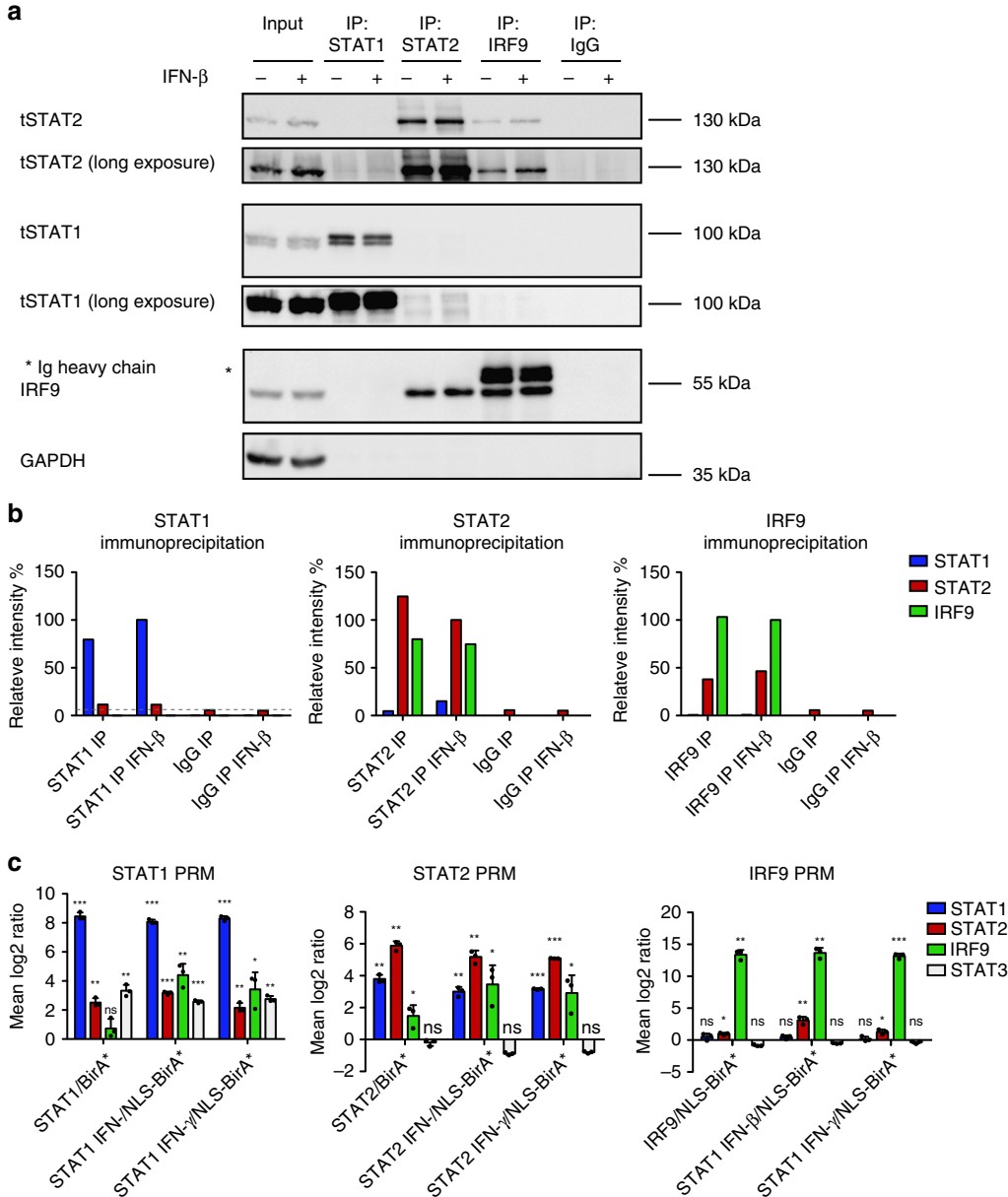

**Fig. 4** Signal dependence of complex formation from ISGF3 subunits. **a** BMDMs from wild-type (WT) animals were treated for 1.5 h with IFN-β and immunoprecipitation was carried out using antibodies against STAT1, STAT2, IRF9, or an IgG control. Immunoprecipitated complexes were analyzed by western blotting with antibodies to STAT1, STAT2, IRF9, and GAPDH. Input controls represent 10% of the total lysate used for the immunoprecipitation. **b** The representative blot was quantified using ImageJ software. Relative intensities of the bands were normalized to their corresponding input sample. Data represent relative intensities in percent, where STAT1, STAT2, and IRF9 levels in IFN-β-treated IPs equal 100%. **c** Targeted quantitative MS analysis of STAT1, STAT2, and IRF9 BioIDs using PRM. Raw 264.7 cells were treated with 0.2 μg/ml doxycycline for 24 h, followed by addition of 50 μM biotin for 18 h. Cells were treated for 2 h either with IFN-β or IFN-γ lysed and protein complexes were isolated by streptavidin affinity purification, followed by analysis with LC–MS. Mean log$_2$-transformed protein ratios were calculated for three biological replicates of myc-STAT1-BirA*, myc-STAT2-BirA*, or myc-IRF9-BirA* cells normalized to their appropriate localization control. Standard deviation and $t$-test statistics were calculated for each of the target proteins. $P$-values (*$P \leq 0.05$; **$P \leq 0.01$; ***$P \leq 0.001$). Source data are provided as a source data file

endogenous levels of ISGF3 subunits (Supplementary Fig. 5a, c, e) and used cells expressing either N-terminally myc-tagged BirA* or NLS-BirA* genes as controls. The NLS-tagged protein localized to both the cytoplasm and nucleus, whereas the protein without NLS was almost exclusively cytoplasmic (Supplementary Fig. 6a, b). As will be shown below, the BirA* control reflected the subcellular localization of STATs 1 and 2 in resting cells, whereas the localization of NLS-BirA* matched that of IRF9 in both resting and IFN-treated cells and that of the STATs in IFN-treated cells. The two controls were used in accordance with this to normalize our

BioID experiments. We first examined the proximity of ISGF3 subunits prior to IFN treatment by parallel-reaction monitoring (PRM), an approach that allowed to specifically acquire information about peptides of interest and their quantities[29]. Triplicate PRM samples revealed that IRF9-BirA* biotinylated both itself and STAT2 (Fig. 4c, Supplementary Data 6c). In contrast, STAT1 was not enriched by Streptavidin-mediated affinity purification compared with the control cells. In IRF9-BirA* cells, a 2-h pulse with either IFN-I or with IFN-γ did not cause detectable STAT1–IRF9 proximity. Interaction with STAT2

on the other hand increased under these conditions. STAT1-BirA* cells corroborated the lack of STAT1–IRF9 complexes in resting cells and confirmed interaction with STAT2, in line with the co-IP approach. Notably however, STAT1-BirA* revealed proximity to both IRF9 and STAT2 after treatment with either IFN-β or IFN-γ. Having shown that the IRF9-BirA* construct is active in biotinylating STAT2, the discrepancy to the results with STAT1-BirA* must result from steric constraints. The published model of the DNA-ISGF3 complex does not offer an explanation because it proposes proximity of the IRF9 N terminus and the STAT1 DBD[21]. On the other hand, a structural model of the STAT4 N terminus and the dimer formed by the STAT1 core suggests that the two are separated by a flexible stretch of amino acids that allows for the tetramerization of adjacent dimers[30]. Thus, whereas the inability of BirA* fused to the IRF9 N terminus to biotinylate STAT1 cannot be explained, IRF9 biotinylation by STAT1-BirA* is in agreement with structural models.

PRM with STAT2-BirA* confirmed proximity to both IRF9 and STAT1 in resting cells (Fig. 4c). Treatment with IFN further increased the interaction between STAT2 and IRF9.

To demonstrate the specificity of the PRM data, STAT3 was used as an additional control (Fig. 4c). STAT3 is known to form heterodimers with STAT1, but not STAT2 or IRF9 and STAT1: STAT3 heterodimers impact on type I-IFN responsiveness[31,32]. In accordance with this, STAT1-BirA*, but not STAT2-BirA* or IRF9-BirA* reported proximity to STAT3. As in the case of STAT1–STAT2 proximity, biotinylation of STAT3 by STAT1-BirA* was largely unaffected by IFN treatment. Proximity labeling at a later time point (18 h) after IFN treatment did not change the conclusions reached in experiments with a brief IFN pulse (Supplementary Fig. 6c).

Together, the co-IP experiments and the BioID approach concurred in identifying the presence of STAT2–IRF9 and of STAT1–STAT2 heterodimers in resting cells. Whereas our co-IP experiments and findings in the literature[5,33] consistently excluded the formation of stable ISGF3 complexes with both STAT1 and IRF9 subunits in the absence of DNA, the experiments with STAT1-BirA* cells demonstrate proximity of these subunits after IFN treatment. A likely explanation is that the BioID experiment detects STAT1–IRF9 proximity when the complete ISGF3 complex forms on DNA in IFN-treated cells. This assumption is consistent with data in Fig. 2 and predicts that DNA-mediated precipitation should detect a complete ISGF3 complex in vitro. To verify this assumption, we carried out DNA-mediated precipitation studies (Fig. 5a). Oligonucleotides representing Isg15 or Oas1a ISREs, i.e., binding sites that show a STAT2–IRF9 to ISGF3 shift in Fig. 2, precipitated STAT2–IRF9 from resting-cell extracts, but the entire ISGF3 complex only after stimulation with IFN-I (the weak STAT1 band in resting-cell extracts in lane 1 was also observed in the absence of specific ISRE DNA in lanes 5 and 6 and results from nonspecific association to the carrier material). We conclude that IRF9 biotinylation in STAT1-BirA cells reflected DNA-associated ISGF3 complexes and that ISRE binding is required for the formation of stable ISGF3 complexes.

In addition to the PRM analysis shown above, we determined the ISGF3 interactome according to BioID (Fig. 5b, Supplementary Data 7 shows raw data for the complete list of interactors). The overlap of proteins showing proximity to all three ISGF3 subunits was low, but about 50% interacted with more than one subunit across one experimental condition. Most likely, this reflects the occurrence of the subunits in distinct molecular complexes and/or steric constraints of the BirA* domains in the biotinylation reaction. IFN treatment altered the interactome, but the majority of interactions was constitutive (Fig. 5b, Supplementary Data 7). As expected, functional annotation highlighted proteins involved in transcription control or chromatin structure/

modification as one category (Supplementary Data 8). Proteins participating in the DNA damage response were also found. Among these, PARP14 was recently shown to be required for the nuclear accumulation of IFN-induced proteins and for transcriptional activation of IRF3 target genes, i.e., ISRE-based transcriptional activation[34]. ISGF3 interactors encoded by ISGs included Pyhin family members that are also interactors of the DNA damage response pathway[35]. Surprisingly, the largest functional group were proteins involved in cell metabolism, including a considerable number of mitochondrial enzymes and a subunit of the outer mitochondrial membrane transporter, TOMM70. STAT1 reportedly localizes to mitochondria and impacts on mitochondrial metabolism[36], and small quantities of STAT2 were found in mitochondria as well[37]. These reports support the notion that the large number of mitochondrial interactors indeed reflect a mitochondrial localization of ISGF3 subunits.

**Signaling requirements for STAT2–IRF9 formation**. The revving-up model of innate immunity predicts that the molecular machinery for constitutive ISG expression requires a low chronic signal from the IFN receptor. It further implies that a small quantity of ISGF3 subunits should be nuclear in untreated cells. To test these assumptions, we first analyzed the cellular localization of ISGF3 subunits. We have recently shown that IFN-induced nuclear localization of STAT2 is reduced in Irf9[-/-] BMDM[38], supporting the conclusion that nuclear shuttling of at least a subfraction of STAT2 requires IRF9. On the other hand, human IRF9 was shown to contain a nuclear retention signal in the DNA-binding domain and high levels of STAT2 retained part of IRF9 in the cytoplasm[39,40]. Together, the data demonstrate a mutual impact of STAT2 and IRF9 on each other's subcellular localization. In mouse BMDM, immunofluorescence localized a large part of IRF9 to the nucleus, both before and after IFN-I treatment (Fig. 6a). To examine the signaling dependence of the cytoplasm/nucleus distribution of STATs, we examined BMDM in the presence and absence of the pan-JAK inhibitor P6 at a concentration that abrogated IFN-induced STAT1/2 tyrosine phosphorylation (Fig. 6b, c, f). Cell fractionation and western blots revealed that nuclei from resting cells contained IRF9 and a small fraction of STAT2, but not STAT1 (Fig. 6b–d). Evidently, the small quantity of STAT1 bound to chromatin according to ChIP-seq is below the detection limit. In the presence of P6, nuclear STAT2 remained largely unchanged, suggesting that it is independent from a receptor signal (Fig. 6c, e). In contrast, IFN-induced nuclear translocation of STAT2 was abrogated by JAK inhibition. With the exception of a signal-dependent increase in nuclear presence after IFN-β treatment, the cytoplasm/nucleus distribution of IRF9 was unaffected by JAK inhibition (Fig. 6d, f). Taken together, these observations support the conclusion that the formation and nuclear localization of STAT2–IRF9 complexes occur independently of a continuous signal from the IFN-I receptor.

**Tonic IFN signaling maintains expression of ISGF3 subunits**. The permanent presence of STAT2–IRF9 in the nucleus suggests that basal ISG expression should be largely unaffected by JAK inhibition. To test this hypothesis, untreated or P6-treated WT BMDM were compared with the ISGF3-subunit knockouts (knockout data are as in Fig. 1 and included here for ease of comparison). In agreement with our assumption, basal expression of genes with pre-bound STAT2–IRF9 was largely maintained in the presence of inhibitors (Irf7, Usp18, and Oas1a; Fig. 7a). The same concentrations of JAK inhibitor strongly reduced the induction of these genes by IFN-β (Supplementary Fig. 7a). Contrasting with inhibitor treatment, knockouts of all three

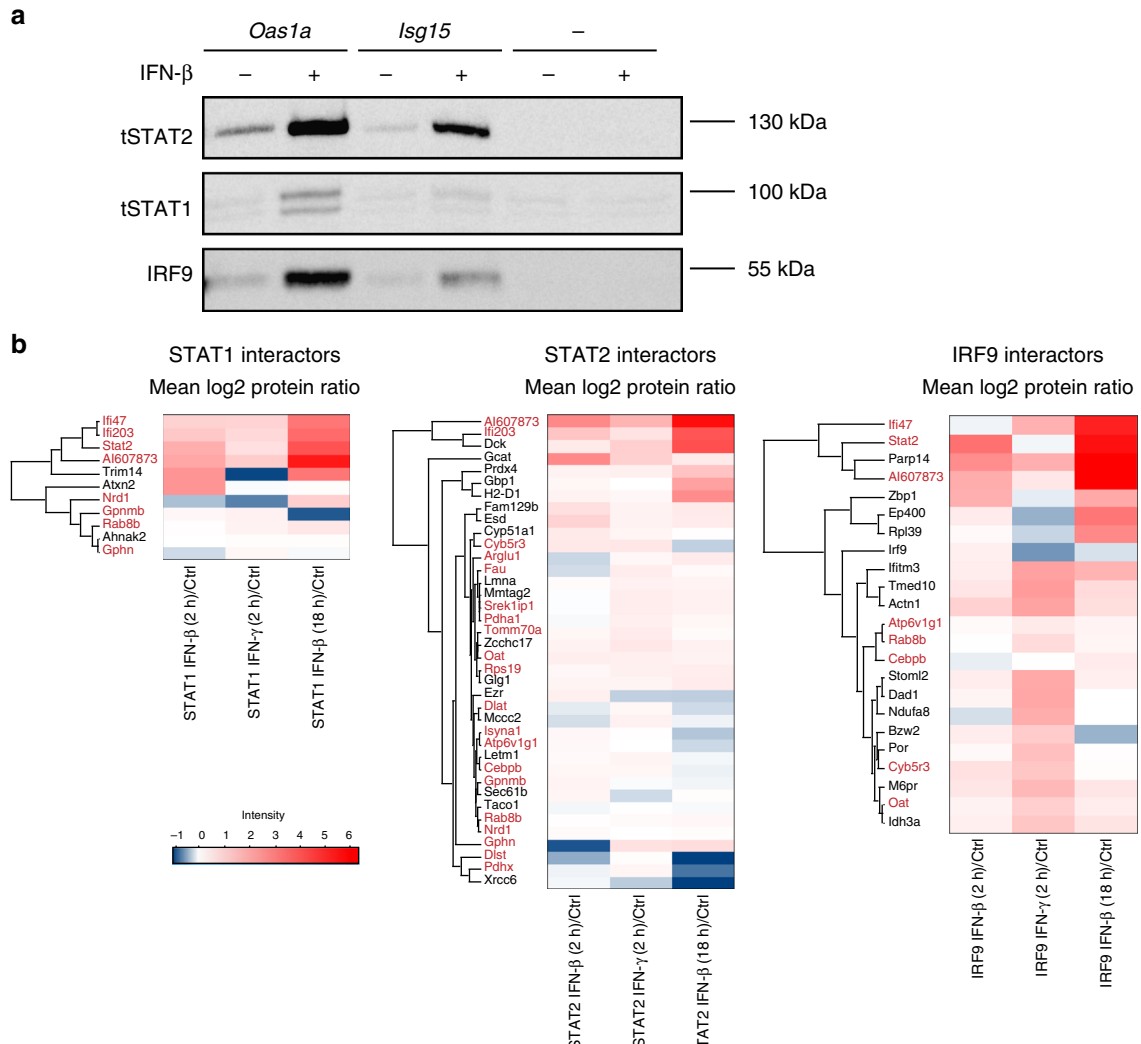

**Fig. 5** Complex formation of ISGF3 subunits and proximity labeling of interactors. **a** Raw 264.7 cells were treated for 1.5 h with IFN-β. Cell lysates were incubated with a biotinylated Oas1a-ISRE oligo, a biotinylated Isg15-ISRE oligo, or plasmid DNA. DNA-bound protein complexes were isolated by streptavidin affinity purification, followed by western blot analysis. **b** STAT1, STAT2, and IRF9 interactome dynamics in response to interferon treatment. Hierarchical cluster analysis of proteins significantly enriched upon treatment with IFN-β or IFN-γ. Proteins were filtered, which were at least twofold enriched above background (myc-BirA* or BirA*-NLS controls) in at least one condition at an adjusted $p$-value of < 0.01, and which showed at least a twofold increase in intensity after interferon induction when compared with steady-state conditions. For this filtered set of proteins, we computed the mean log$_2$ LFQ protein ratio of the interferon-induced (2 and 18 h) and the steady-state condition and used these values to generate a hierarchical cluster analysis and heat maps in Perseus with default settings. Interactor names shown in red were found associated with more than one ISGF3 subunit across all experimental conditions, including interactors under resting conditions shown in Supplementary Data 7. Source data are provided as a source data file

ISGF3 subunits strongly affected STAT2–IRF9-dependent genes, whereas genes induced via GAS sequences (*Irf1*, *Irf8*) were unaffected by either the inhibitor or the knockouts (Fig. 7a). Thus, shutting down signaling while maintaining ISGF3-subunit levels, as is the case in our inhibitor experiments, sustains basal ISG expression independent of tonic IFNAR signaling.

The persistence of nuclear STAT2–IRF9 upon JAK inhibition raises the question of why genes whose basal expression is sustained by this complex are sensitive to a permanent disruption of signaling in cells lacking the IFN-I receptor or STAT1, or that express a STAT1Y701F mutant[38]. A straightforward answer to this question is provided by the results in Figs. 3e and 7b showing that ISGF3 subunits are bound to the promoters of each of their genes in resting MEFs and BMDM, suggesting that they all contribute to each other's basal expression. Consistent with this, *Stat*1 and *Irf9* promoters were associated with ISGF3 and the

*Stat2* promoter bound STAT2–IRF9. Thus, gene deletion of all three ISGF3 subunits is expected to lower IRF9 levels and therefore any ISRE-dependent basal expression. Consistently, knockout of each subunit caused a severe reduction of the two other subunits in both macrophages (Fig. 7c) and fibroblasts (Fig. 7d). Thus, cells lacking the ability to form an ISGF3 complex express low amounts of all its subunits and are therefore unable to sustain STAT2–IRF9-dependent basal gene expression. Consistent with this notion, the introduction of a Dox-inducible Stat1 transgene into *Stat1*[−/−] MEFs restored the expression of both STAT2 and IRF9 (Fig. 7e, f). The data emphasize the importance of studying STAT complex formation in wt conditions. An integrated model depicting the proposed interplay between tonic signaling-dependent and independent events for basal ISG expression and the changes occurring upon IFN treatment are depicted in Fig. 8.

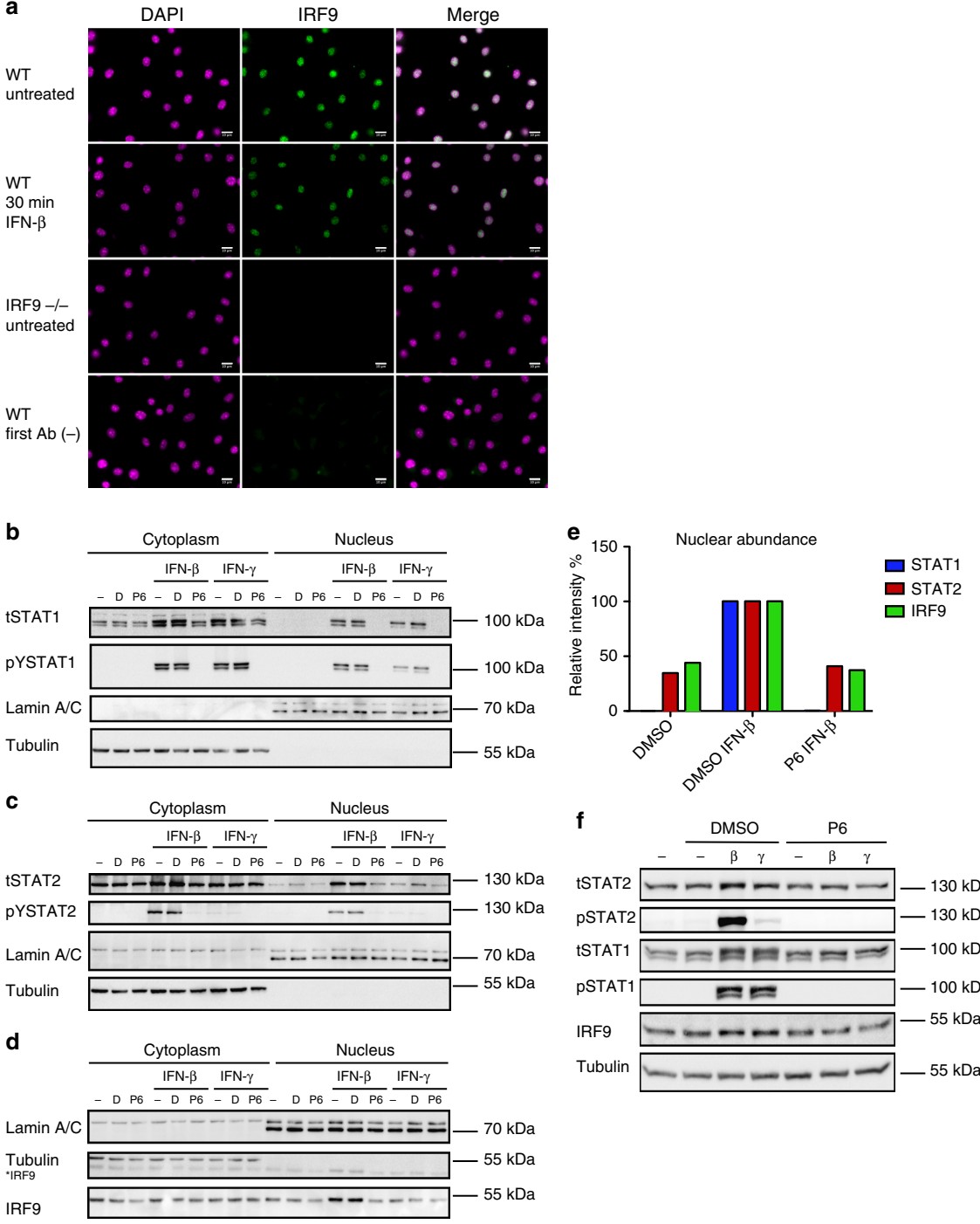

**Fig. 6** Localization of STAT complexes in BMDM. **a** IRF9 localization as determined by immunofluorescence. BMDMs of wild-type (WT) and *Irf9*⁻/⁻ (*IRF9*⁻/⁻) mice were left untreated or stimulated with IFN-β for 30 min. The cells were fixed and stained with an anti-IRF9 antibody followed by Alexa Fluor® 488-conjugated secondary antibody (green). Nuclei were stained with DAPI (magenta). First Ab (−) indicates the control without the first antibody. The scale bars represent 10 μm. **b–d** Nuclear and cytoplasmic extracts from BMDM were prepared from controls or after a 30-min treatment with IFN-β or IFN-γ and analyzed by western blot. A 2:1 ratio of the nuclear-to-cytoplasmic fraction is shown. Where indicated, 15 μM P6 inhibitor or DMSO were added for 3 h prior to IFN treatment. Phosphorylation of STAT1 at Y701 and total STAT1 levels, as well as phosphorylation of STAT2 at Y689, total STAT2 and IRF9, and α-tubulin and lamin A/C levels were determined. **e** The nuclear fractions of the representative blots **b–d** were quantified using ImageJ software. Relative intensities of the bands were normalized to their corresponding lamin C levels. Data represent relative intensities in percent, where STAT1, STAT2, and IRF9 levels in IFN-β-treated nuclear extracts equal 100%. **f** Phosphorylation of STAT1 at Y701, STAT2 at Y689, total STAT1, total STAT2, IRF9, and α-tubulin was determined in whole-cell lysates of BMDM. Overall, 15 μM P6 inhibitor or DMSO were added for 3 h prior to IFN treatment. Source data are provided as a source data file

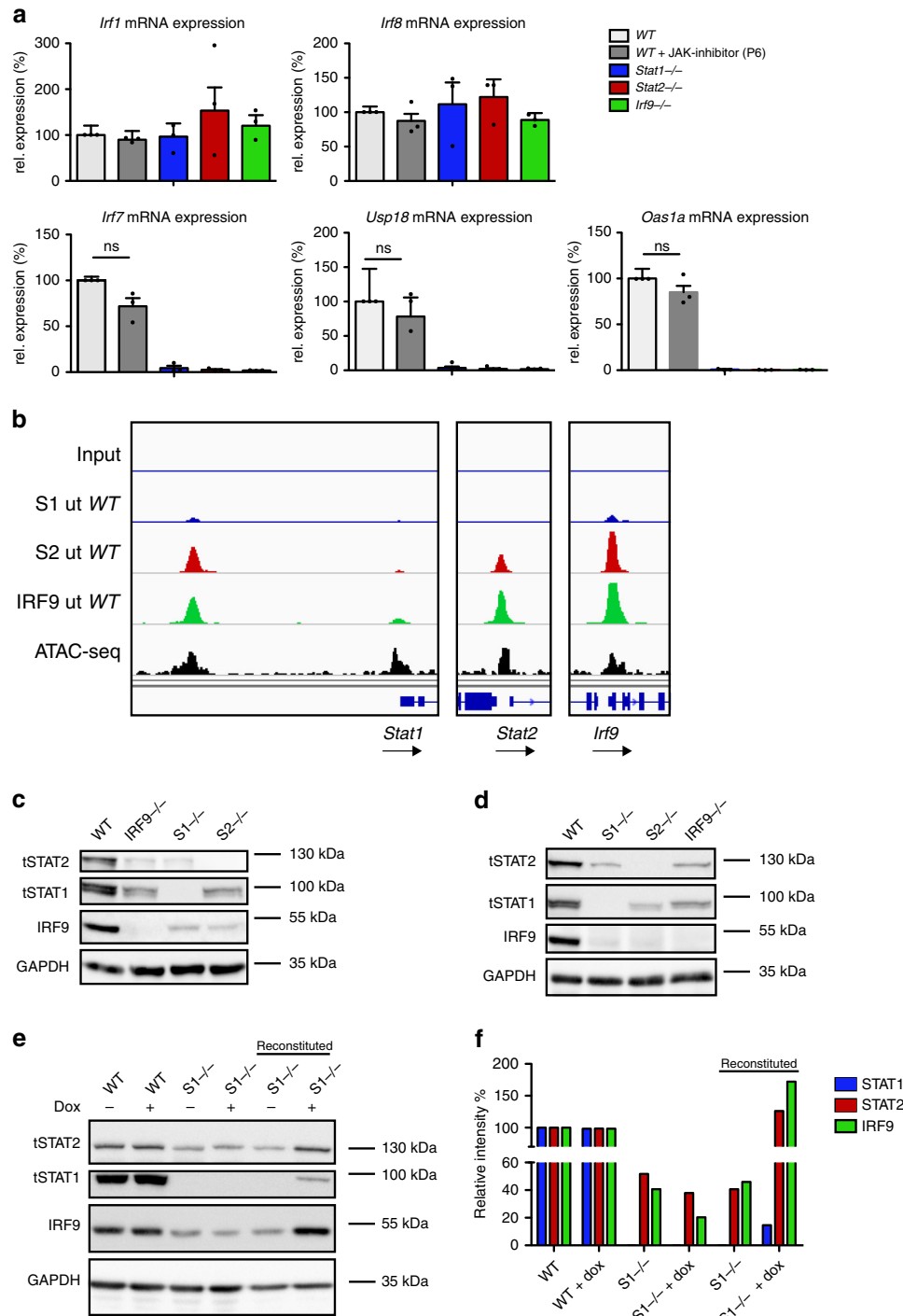

**Fig. 7** Cross-regulation of ISGF3 subunits. **a** BMDMs isolated from WT, $Stat1^{-/-}$, $Stat2^{-/-}$, and $Irf9^{-/-}$ mice were left untreated or treated with 15 µM P6 inhibitor for 3 h. Gapdh-normalized gene expression was measured by RT-q-PCR. Data represent relative expression in percent, where WT untreated equals 100%. Data represent the mean and standard error of the mean (SEM) values of three independent experiments. $P$-values were calculated using the paired ratio $t$-test (*$P \leq 0.05$; **$P \leq 0.01$; ***$P \leq 0.001$). **b** STAT1 (S1), STAT2 (S2), and IRF9 binding at the $Stat1$, $Stat2$, and $Irf9$ promoters (scale 0–150). The browser tracks represent data derived from the ChIP-seq experiments in BMDM described in the legend of Fig. 2. **c** Whole-cell extracts from wild-type, $Stat1^{-/-}$, $Stat2^{-/-}$, and $Irf9^{-/-}$ BMDMs were tested by western blot for total STAT1, STAT2, and IRF9 levels. **d** Whole-cell extracts from wild-type, $Stat1^{-/-}$, $Stat2^{-/-}$, and $Irf9^{-/-}$ mouse embryonic fibroblasts were analyzed by western blot for total STAT1, STAT2, and IRF9 levels. **e** $Stat1^{-/-}$ mouse embryonic fibroblasts were stably transduced with a doxycycline-inducible STAT1-myc construct. Whole-cell extracts from wild-type, $Stat1^{-/-}$, and reconstituted $Stat1^{-/-}$ MEFs were tested by western blot for total STAT1, STAT2, and IRF9 levels in the absence and presence of doxycycline. **f** The representative blot in **e** was quantified using ImageJ. Relative intensities of the bands were normalized to their corresponding GAPDH levels. Data represent relative intensities in percent, where STAT1, STAT2, and IRF9 levels in untreated wild-type MEFs equal 100%. Source data are provided as a source data file

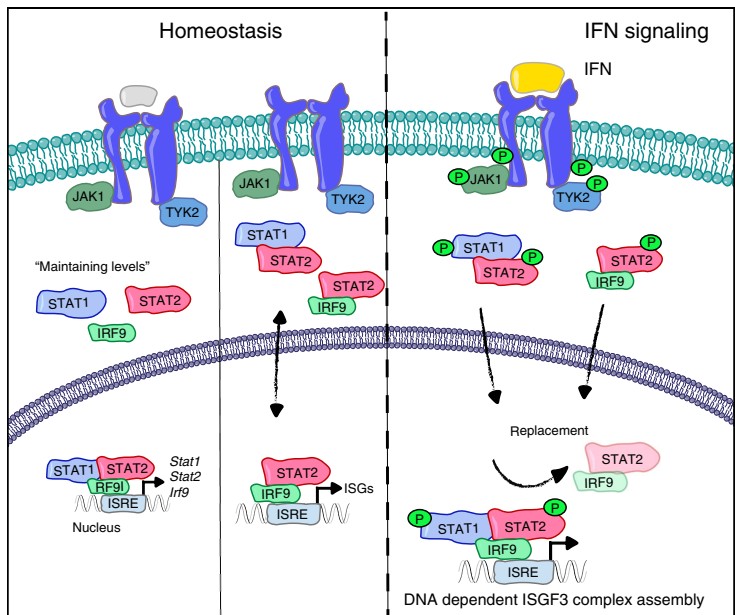

**Fig. 8** Model of the molecular switch from resting to IFN-induced gene expression. Under homeostatic conditions, a tonic signal from the type I-IFN receptor activates small quantities of ISGF3, which increase basal expression of the genes encoding the ISGF3 subunits STAT1, STAT2, and IRF9. This causes the constitutive formation of STAT1–STAT2 as well as STAT2–IRF9 complexes. Basal expression of a large fraction of interferon-induced genes (ISGs) is stimulated by STAT2–IRF9 complexes that appear in the nucleus without a signaling requirement. Signaling by the type I-IFN receptor and to a significant extent also by the IFN-γ receptor causes the formation of tyrosine-phosphorylated STAT1–STAT2 heterodimers that translocate to the nucleus and form an ISGF3 complex by associating on DNA with IRF9. The higher off-rate of STAT2–IRF9 compared with ISGF3 combined with a larger quantity of tyrosine-phosphorylated STAT1–STAT2 versus STAT2–RF9 complexes in IFN-treated cells most likely explains why a rapid exchange takes place after interferon treatment

## Discussion

Recent findings suggest that unphosphorylated STAT complexes control ISG expression[11]. Our study shows an important contribution of STAT2–IRF9 complexes, formed independently of IFN-I receptor signaling, to constitutive ISG expression in resting cells. This added mechanistic insight calls for a reinterpretation of the "revving-up" model and the implications of tonic signaling by the IFN-I receptor[15,17]. Both models provide a mechanistic framework to explain how cells are prepared for the race between pathogen multiplication and innate resistance. They posit that IFN-dependent responses, such as the antiviral state or macrophage activation, are not established de novo during immune responses, but represent the enhancement of a pre-existing condition compatible with resting-state physiology. We confirm that tonic signaling is indeed of critical importance for the maintenance of ISGF3 subunits and show that the signal-independent formation of STAT2–IRF9 complexes, previously considered as noncanonical, is an integral component of ISG regulation. The change from STAT2–IRF9 to ISGF3 functions as a molecular switch between resting and active states for many ISGs. Importantly, our observations rely on wt cells. Our data with knockout cells show that deletion of one STAT subunit not only extinguishes the function of that protein, but also changes the stoichiometry of the remaining ISGF3 components. Component stoichiometry is an important determinant for the proper biological output of JAK-STAT signaling. For example, IL-6 signaling in the absence of STAT3 exhibits a transcriptional profile similar to that of IFN-γ by preferentially using STAT1[41].

Basal ISG expression requires the constant secretion of miniscule IFN-I amounts to prime cells for a rapid response to microbial infections. This is illustrated by IFN-β promoter–luciferase reporter mice that show weak luminescence in the entire body[42]. Earlier studies revealed IFN-I mRNA and protein in tissues under pathogen-free conditions[43,44]. Besides priming cells for enhanced

cytokine responsiveness, this homeostatic mechanism reveals its importance in the maintenance and mobilization of the hematopoietic stem cell niche[45,46]. More recent studies suggest that multipotent stem cells achieve ISG expression and an antiviral state independently of IFNAR signaling[47]. This result strengthens the idea that mobilization of ISG promoters via STATs can be uncoupled from IFN-I receptor signaling. The uSTAT model provided a first proof of principle for this possibility. Accordingly, the prolongation of ISG expression occurs via formation of a u-ISGF3 complex following an early, IFNAR-dependent response that increases the levels of ISGF3 subunits[12]. In hepatic cells, a u-ISGF3 was proposed to function under homeostatic conditions and without a previous IFNAR-dependent signal[48]. However, a mechanism explaining IFNAR-independent ISG expression under conditions of a wild-type cell remained elusive. Here, we show that the switch of many ISGs from basal to rapid IFN-induced expression requires exchanging a largely signal-independent STAT2–IRF9 for ISGF3 complexes. This original observation in BMDM was confirmed in MEFs. In THP-1 cells, a group of ISGs required IRF9 for basal expression, but in line with a previous study in HeLa cells[49], did not pre-associate with STAT2–IRF9 according to ChIP-seq. This apparent contradiction with the IRF9 dependence of expression may result from technical limitations, an inferiority of the reagents for ChIP of the human ISGF3 subunits compared with those for mouse. Alternatively, it may reflect a difference between the mouse cells analyzed in our study and human monocytes. Our model involving STAT2 and IRF9 in basal ISG expression receives strong support from a report comparing basal gene expression in $Ifnar1^{-/-}$ and $Tyk2^{-/-}$ murine B cells, both lacking tonic signaling[16]. This revealed a large overlap with genes whose expression was impaired by the loss of IRF9 in our study. Importantly, tonic-sensitive loci showed higher STAT2 binding at baseline[16]. Another study in human hepatoma cells also demonstrated constitutive STAT2 association with ISG promoters[50]. Strong support for the

role of IRF9–STAT2 in basal gene expression is also provided by the finding that an IRF9 fusion protein with the STAT2 transactivating domain is sufficient to increase basal ISG expression and to induce an antiviral response in the absence of added IFN[51]. We were surprised about the large proportion of IFN-γ-induced genes affected by IRF9 deficiency and that many of the STAT2–IRF9 to ISGF3 switching promoters responded to IFN-γ. Consistent with this, the BioID approach showed proximity of all subunits in IFN-γ-treated macrophages. While the fraction of genes associated with STAT1 dimers was clearly larger than in IFN-β-treated cells, our data emphasize the importance of the ISGF3 complex for transcriptional responses to IFN-γ. The large overlap between type I-IFN-induced and IFN-γ-induced ISGs most likely explains common activities of all IFN types, such as the antiviral state.

Earlier studies showing pre-association of STAT2 and IRF9 proposed cytoplasmic localization of the complex, due to a dominant STAT2 nuclear export sequence[39,40]. Our data revise this idea by showing that a proportion of STAT2–IRF9 resides in the nucleus of resting cells, most likely because it is trapped on DNA. Importantly, our antibody- and DNA-mediated co-precipitation studies argue in favor of an ISGF3 formation on DNA. They are consistent with earlier biochemical work demonstrating disintegration of a purified ISGF3 complex in the absence of DNA[33] and with a study concluding added stability of the complex in the presence of a DNA-binding site[5]. In the cell nucleus, the switch to ISGF3 might be affected by the addition of a STAT1 dimer to a pre-associated STAT2–IRF9 complex. However, this model contradicts the proposed 1:1:1 stoichiometry of ISGF3 subunits[21,33]. In the light of available evidence, the most likely scenario is that the pre-associated STAT2–IRF9 is replaced by an ISGF3 complex that forms on the ISRE from tyrosine-phosphorylated STAT1–STAT2 heterodimers and IRF9. One reason for the exchange and also the transcriptional increase associated with ISGF3 binding is most likely the improved stability on DNA. While IRF9 makes contacts to the core ISRE, STAT1 increases ISGF3 affinity by additionally interacting with 5′ flanking sequences[52,53]. The higher off-rate of STAT2–IRF9 in the binding equilibrium with the ISRE sequence measured by Bluyssen and Levy[53] in vitro provides a rationale for the rapid exchange of this transcription factor for ISGF3 after IFN treatment. An additional implication might be transcription factor cooperativity. For example, a STAT2–IRF9 complex reportedly allows for co-regulation of the human IL-6 gene by IFN and NFκB-activating agents, because STAT2 binds IRF9 as well as the p65 subunit of NFκB[54]. Consistently, data by Mariani et al. suggest that STAT2–IRF9 mediates cooperativity required for the enhanced induction of genes in response to IFN-β and the NFκB-activating cytokine TNF[55]. Thus, it is possible that STAT2–IRF9 and ISGF3 represent alternative platforms for the integration of transcription factor signaling at the promoter level. In vivo biotinylation shows ISGF3 subunits to be in close proximity to a large number of proteins involved in chromatin organization and transcription initiation. This is consistent with IFN signal-dependent alterations of chromatin modification and structure, including the exchange of histones[16,49,56]. EP400, a subunit of the Tip60/NuA4 chromatin remodeling complex[57] associated with histone exchange[58], was found in proximity to IRF9. Additional components of this complex, such as the ATP-binding proteins RVB1 and RVB2 interact with the transactivation domain of STAT2 in the nuclei of IFN-stimulated cells and are required for robust ISG activation by type I-IFN[59]. In our screen. RVB2 (i.e., mouse Ruvbl2) was among proteins biotinylated by STAT2-BirA with a padj value of <0.01, but with a log-FC above background of 0.8, whereas our cutoff was 1. EP400 is a scarce example of an ISGF3 interactor with clear functional implications. The consequences of most other interactions, such as with the

aforementioned DNA damage response proteins or metabolic enzymes, cannot be interpreted with the current state of knowledge. Thus, the proximity screen reported here opens up a wide field for future investigations. The results with the ISGF3 subunits alone delineate an astounding complexity. An integrated model of IFN-I signaling must ultimately account not only for the STAT2–IRF9 to ISGF3 transition investigated here, but also for the aforementioned u-ISGF3 complexes[12] or the preformed STAT1–STAT2 heterodimers identified by Ho et al. with biochemical approaches[24] and confirmed here in intact cells by proximity labeling. IFN-independent STAT1–STAT2 association may both be relevant for the rapid formation of the tyrosine-phosphorylated heterodimer after IFN treatment[22] and prevent its dissociation after dephosphorylation[60]. As described above, STAT2 and IRF9 mutually influence each other's subcellular localization. Likewise, hemiphosphorylated pSTAT1-STAT2 dimers resulting from the preferential tyrosine phosphorylation of STAT1 dampen the response to IFN-γ by restricting STAT1 access to the nucleus[24]. The multipurpose employment of ISGF3 subunits thus represents a striking example of the cell's economy in the management of complex regulatory tasks.

## Methods

**Animal experiments.** Animal experiments were approved by the institutional ethics and animal welfare committee of the University of Veterinary Medicine, Vienna, and the national authority (Austrian Federal Ministry of Education, Science, and Research) according to §§26ff of Animal Experiments Act (Tierversuchsgesetz TVG 2012, BGBl. I Nr 114/2012) under the permission license numbers BMWF 68.205/0032-WF/II/3b/2014 and BMWFW-68.205/0212-WF/V/3b/2016. Animal husbandry and experimentation was performed under the Austrian national law and the ethics committees of the University of Veterinary Medicine Vienna and according to the guidelines of FELASA, which match those of ARRIVE. C57BL/6 N, Irf9[−/−], Stat1[−/−], and Stat2[−/−] mice[61–63] were backcrossed for more than ten generations on a C57BL/6 N background, were housed in the same specific-pathogen-free (SPF) facility under identical conditions according to recommendations of the Federation of European Laboratory Animal Science Association, and additionally monitored for being norovirus negative.

**Cell culture.** Bone marrow-derived macrophages (BMDM) were differentiated from bone marrow isolated from femurs and tibias of 8- to 12-week-old mice from both sexes. Femur and tibia were flushed with Dulbecco's modified Eagle's medium (DMEM) (Sigma-Aldrich), and cells were cultured in DMEM supplemented with 10% of fetal calf serum (FBS) (Sigma-Aldrich), 10% L929-cell- conditioned medium as a source of colony-stimulating factor 1 (CSF-1), 100 units/ml penicillin, and 100 ng/ml streptomycin (Sigma-Aldrich). Cells were kept at 37 °C and 5% $CO_2$ and differentiated for 10 days. For ChIP-seq and RNA-seq experiments, BMDMs were differentiated in DMEM containing recombinant M-CSF (a kind gift from L. Ziegler-Heitbrock, Helmholtz Center, Munich, Germany). BMDM and Raw 264.7 cells were stimulated with 10 ng/ml murine IFN-γ (eBioscience; Catalog # 14-8311-63) or 250 IU/mL of IFN-β (PBL Assay Science; Catalog # 12400-1).

Human monocytic THP-1 cells (ATCC #TIB-202) were maintained in RPMI 1640 culture medium (Sigma-Aldrich) supplemented with 10% heat-inactivated fetal bovine serum (FBS) (Sigma-Aldrich) and 1% Penicillin/Streptomycin (both Sigma-Aldrich) (here referred to as "complete medium"). Human monocytic THP-1 cells were differentiated into macrophage-like cells by treating them with 100 nM phorbol 12-myristate 13-acetate (PMA) (Sigma-Aldrich, Catalog # P1585) in complete media for 48 h at 37 °C and 5% $CO_2$ atmosphere. On the third day, media containing 100 nM PMA were changed to complete media and incubation at 37 °C and 5% $CO_2$ atmosphere was continued for an additional 24 h.

Raw 264.7 macrophages (ATCC #TIB-71) and mouse embryonic fibroblasts (MEFs) deficient in STAT1, STAT2, or IRF9 were cultured in DMEM (Sigma-Aldrich) supplemented with 10% FBS (Sigma-Aldrich), and with penicillin and streptomycin (Sigma-Aldrich). All immortalized cell lines were routinely tested for mycoplasma contamination.

**Genome editing via CRISPR–Cas9 system.** The guide RNA (ATACAGCTAAG ACCATGTTC (CGG)) of human Irf9 was designed using Broad Institute GPP Web Portal (https://portals.broadinstitute.org/gpp/public/analysis-tools/sgrna-design). Oligos were ligated into LentiCRISPRv2 plasmid (a kind gift from L. Versteeg, MPL) and transduced into THP-1 cell. Single clones were selected.

**Generation of monoclonal mouse IRF9 antibody.** The murine monoclonal anti-IRF9 antibody was generated in collaboration with Egon Ogris, Stefan Schüchner, and Florian Martys from the MPL monoclonal antibody facility. Full-length

murine *Irf9* was cloned into a pET-Duet1 (Novagen, Catalog # 71146) vector and expressed in *E. coli* Rosetta pLysS strain and purified using Ni-sepharose beads. Hybridomas from antibody-producing B cells and myeloma cells for the production of monoclonal IRF9 antibodies were generated. The best signal-to-noise ratio in ChIP and western blot analysis was obtained with the single clone 6F1-H5, which was used for this study. The purified antibody can now be purchased from Sigma (Anti-IRF-9, clone 6F1-H5, Cat. No. MABS1920, EMD Millipore).

**RNA isolation, cDNA synthesis, and q-PCR**. Total RNA was extracted from mouse bone marrow-derived macrophages and MEFs using the NucleoSpin RNA II kit (Macherey-Nagel, Catalog # 740955). The cDNA was prepared using Oligo (dT18) Primer and the RevertAid Reverse Transcriptase (Thermo-Fisher Scientific). Real-time q-PCR experiments were run on the Mastercycler (Eppendorf) to amplify the Gapdh (housekeeping gene), using SybrGreen (Promega). Primers for q-PCR and ChIP q-PCR are listed in Supplementary Table 1.

**Western blot**. Cells were lysed in Laemmli buffer (120 mM Tris-HCl, pH 8, 2% SDS, and 10% glycerol). Protein concentration was determined (Pierce BCA Protein Assay Kit). Thirty micrograms of protein were mixed with β-mercaptoethanol and bromophenol blue, boiled, and loaded on a 10% SDS polyacrylamide gel.

Proteins were blotted on a PVDF or on a nitrocellulose membrane at 4 °C for 16 h at 200 mA and then for 2 h at 400 mA in carbonate transfer buffer (3 mM Na₂CO₃, 10 mM NaHCO₃, and 20% ethanol).

The membrane was blocked in 5% milk powder in TBS-T for 1 h at room temperature. The membrane was washed three times with TBS-T and then incubated with the primary antibody overnight at 4 °C while shaking. Two antibodies against Stat1 were used in experiments with BMDM and MEFs, respectively (Cell Signaling, Catalog # 9172 1:1000 and Santa Cruz, Catalog # sc-346; 1:1000); Stat2 (Cell Signaling, Catalog # 72604, 1:1000); α-Tubulin (Sigma, Catalog # T9026, 1:5000); Phospho-Stat1 (Tyr701; Cell Signaling, Catalog # 9167, 1:1000); Phospho-STAT2 (Tyr689, Merck, Catalog # 07-224, 1:1000); Lamin A/C (Santa Cruz, Catalog # sc-376248, 1:1000); GAPDH (Millipore, Catalog # ABS16, 1:3000); IRF9 (6F1, hybridoma supernatants used for experiments with mouse cells, 1:20); IRF9 (used for experiments with THP-1 cells, Santa Cruz, Catalog # sc-10793, 1:1000); anti-myc (Cell Signaling, Catalog # 2276, 1:1000). The next day, the membrane was washed three times with TBS-T and incubated with an appropriate HRP-coupled secondary antibody for 1 h at room temperature (Jackson ImmunoResearch Inc., Catalog # 111-035-003, 1:6000 and Catalog # 115-035-144, 1:6000). The membrane was analyzed with the ChemiDoc™ Imaging System from Bio-Rad. The western blots were quantified using ImageJ (https://imagej.nih.gov/ij/index.html) under the Gel Analysis Tool. Western blots were saved as png files at 300 dots per inch (d.p.i.). The intensities of the different lanes were taken as a ratio of the protein of interest over the housekeeping gene control and then normalized to the control lane, which was set to 100%. Unprocessed scans of the western blot membranes showing the marker bands, overlayed with the image file of the peroxidase signal, are shown in the Source Data file.

**Nuclear and cytoplasmic extraction**. In total, $5 \times 10^6$ bone marrow-derived macrophage cells were seeded in a 10-cm dish. The next day, cells were treated for 3 h with 15 μM JAK inhibitor (Pyridone 6, Biovision, Catalog # 2534) and afterward stimulated for 30 min either with IFN-β (PBL Assay Science, Catalog # 12400-1, 250 U/ml) or with IFN-γ (eBioscience; Catalog # 14-8311-63, 10 ng/ml). Extraction was carried out according to the manufacturer's instructions (NE-PER™ Nuclear and Cytoplasmic Extraction Reagents, Thermo Fischer, Catalog # 78833). The nuclear and cytoplasmic fraction were loaded in a 2:1 ratio on a 10% SDS gel.

**Immunofluorescence**. In total, $2 \times 10^5$ BMDMs were seeded on glass cover slides, treated with IFN-β for 30 min, and fixed with 3% paraformaldehyde for 20 min at room temperature. Cells were permeabilized with 0.1% saponin in 0.5 M NaCl PBS. Blocking and all the stainings were carried out in 0.1% saponin and 1% BSA in 0.5 M NaCl PBS. The IRF9 antibody was used as a 1:10 dilution overnight at 4 °C. Secondary goat anti-mouse Alexa Fluor® 488 igGH + L (1:500; Catalog # A-11001) was purchased from Invitrogen. Samples were mounted in DAPI (ProLong™ Diamond Antifade Mountant with DAPI, Invitrogen, Catalog # P36962). Images were acquired using Zeiss Axio Imager Z2 with ×63 oil objectives. Images were processed and analyzed using the ImageJ software. The background range in all images for DAPI was adjusted to 1500–16383 and for GFP 3000–16383. The images were changed to 14 bit. A composite picture was made and DAPI was set to magenta and GFP to green. The type of picture was converted into RGB and a scale bar displaying 10 μm was inserted.

**RNA-seq**. In total, $1.5 \times 10^7$ BMDMs, PMA-treated THP-1 cells, as well as mouse embryonic fibroblasts were seeded on 15-cm dishes. The next day, cells were stimulated for 2 h either with 250 U/ml of IFN-β (PBL Assay Science; Catalog # 12400-1 (murine) or PBL Assay Science; Catalog # 11420-1 (human)) or with 10 ng/ml IFN-γ (eBioscience; Catalog # 14-8311-63). Seven milliliters of Qiazol Lysis Reagent (Qiagen) were added per 15-cm dish. Cells were scraped and vortexed for 20 s. In total, 1 ml of the RNA samples in Trizol were used for the RNA prep and 200 μl of chloroform were added. Samples were vortexed for 15 s and

centrifuged for 5 min at full speed at room temperature. The supernatant was mixed with 1 volume of isopropanol, as well as 1/10 volume of 5 M NaCl were added. Samples were incubated for 10 min at room temperature and centrifuged at 16,000 g for 30 min at 4 °C. The pellets were washed twice with 75% EtOH, dried, and resuspended in 30 μl of dH₂O.

For DNase treatment and cleanup, the RNase-free DNase Set (Qiagen, Catalog # 79254) and RNeasy Mini Kit (Qiagen, Catalog # 74104) were used. For library preparation, the NEBNext® Poly(A) mRNA Magnetic Isolation Module together with the NEBNext Ultra II RNA Library Prep Kit from NEB Catalog # E7770L was used according to the manufacturer's protocol. The samples were quality checked and sequenced at the Vienna Biocenter Core Facilities NGS Unit. The RNA-seq experiments were carried out as three independent biological replicates.

**RNA-seq analysis**. Reads mapping to mouse rRNA transcripts were removed using bwa/0.7.12 alignment[64] and samtools/1.3.1[65,66]. The remaining reads were aligned to Mus musculus genome mm10 or human genome hg38 using TopHat v2.1.1[67] (GTF annotation file mm10, RefSeq from UCSC, 2015/02, and hg38 RefSeq from UCSC, 2015/01). Reads in genes were counted with htseq-count v0.6.1.[68] Differential expression analysis for BMDMs was carried out using DESeq 2 v1.16.1[69], with an fdr threshold of 0.05. For comparison between BMDMs and MEFs, as well as BMDMs and THP-1 genes with a minimal lfc ≥ 1 expression and an adjusted *p*-value ≤ 0.05 were considered differentially expressed. Gene set enrichment analysis was performed using GSEA 3.0 against MsigDB v6.1 with gene abundance estimates in FPKM calculated using cufflinks v2.2.1.[70]

**Cloning**. Full-length *Stat1* (NM_001205313.1), *Stat2* (NM_019963.2), and *Irf9* (NM_001159417.1) mouse cDNA were cloned into the pcDNA3.1 mycBioID vector (provided by Kyle Roux) and further subcloned into the pCW57.1 (Catalog # 41393) or pLVX-TRE3G-ZsGreen1 (Catalog # 631350) and used for lentiviral transduction of Raw 264.7 cells. As a negative control for the BioID screen, the BirA*-myc sequence (provided by Kyle Roux; Addgene plasmid Catalog # 35700) and a myc-BirA* carrying an additional NLS sequence were subcloned into the lentiviral pCW57.1 (Catalog # 41393) construct and transduced into Raw 267.4 cells. Murine, N-terminally tagged full-length myc-Stat1α was subcloned into the pCW57.1 (Addgene Catalog # 41393) vector and used for lentiviral transduction of *Stat1*⁻/⁻ MEFs.

**BioID**. BioID was performed according to a published protocol[71] in three biological replicates per construct and condition. *mycBioID* was a gift from Kyle Roux (Addgene plasmid 35700). In total, $5 \times 10^6$ stable Raw 264.7 cells were seeded on 15-cm dishes and treated with 0.2 μg/ml doxycycline for 24 h. Overall, 50 μM biotin was added for 18 additional hours. Cells were stimulated for 2 or 18 h with IFN-β (PBL Assay Science; Catalog # 12400-1), and for 2 h with murine IFN-γ (eBioscience; Catalog # 14-8311-63). Cells were washed and lysed at room temperature (lysis buffer: 50 mM Tris, pH 7.4; NaCl 500 mM; 0.2% SDS; EDTA 5 mM + 1× protease inhibitors). Triton X-100 and 50 mM Tris, pH 7.4, were added, and the protein lysates were sonicated 2× for 30 s. Lysates were centrifuged for 5 min at full speed and the supernatants were transferred to a new tube. Magnetic Pierce Streptavidin beads (Catalog # 88817) were washed 3× with lysis buffer. One hundred and five microliters of beads were incubated with 1.3 mg of protein lysate overnight at 4 °C. Twenty-one microliters of beads were kept for western blot analysis; the rest of the beads were used for the analysis with liquid chromatography–mass spectrometry. Beads were washed at room temperature with wash buffer 1 (2% SDS in H₂O), wash buffer 2 (0.1% deoxycholic acid; 1% Triton X-100, 1 mM EDTA, 500 mM NaCl, and 50 mM HEPES), and wash buffer 3 (0.5% deoxycholic acid, 0.5% NP-40, 1 mM EDTA, 250 mM LiCl, and 10 mM Tris, pH 7.4).

Beads were washed five times with 50 mM Tris, pH 7.4, two times with 50 mM ammonium bicarbonate (ABC), and then resuspended in 24 μL of 1 M urea in 50 mM ABC. Overall, 10 mM dithiothreitol (DTT) was added and the samples were incubated for 30 min at room temperature before adding 20 mM iodoacetamide and incubating for another 30 min at room temperature in the dark. The remaining iodoacetamide was quenched by adding 5 mM DTT and the proteins were digested with 300 ng (Trypsin Gold, Promega) at 37 °C overnight. After stopping the digest by addition of 0.5% trifluoroacetic acid (TFA), and washing the beads with 30 μL of 0.1% TFA, the supernatants were loaded onto C18 stagetips to desalt the peptides prior to LC–MS[72].

Peptides were separated on an Ultimate 3000 RSLC nano-flow chromatography system (Thermo-Fisher), using a pre-column for sample loading (Acclaim PepMap C18, 2 cm × 0.1 mm, 5 μm, Thermo-Fisher), and a C18 analytical column (Acclaim PepMap C18, 50 cm × 0.75 mm, 2 μm, Thermo-Fisher), applying a segmented linear gradient from 2 to 35% solvent B (80% acetonitrile, 0.1% formic acid; solvent A 0.1% formic acid) at a flow rate of 230 nL/min over 120 min for shotgun acquisition and 60 min for targeted acquisition. Eluting peptides were analyzed on a Q Exactive HF X Orbitrap mass spectrometer (Thermo-Fisher), which was coupled to the column with a customized nano-spray EASY-Spray ion-source (Thermo-Fisher) using coated emitter tips (New Objective).

**Shotgun mass spectrometry data acquisition and processing**. The mass spectrometer was operated in data-dependent mode; survey scans were obtained in a mass range of 375–1500 $m/z$ with lock mass activated, at a resolution of 120 k at 200 $m/z$ and an AGC target value of 3E6. The eight most intense ions were selected with an isolation width of 1.6 $m/z$ with 0.2 $m/z$ offset, fragmented in the HCD cell at 28% collision energy, and the spectra were recorded at a target value of 1E5 with a maximum injection time of 250 ms at a resolution of 30k. Peptides with unassigned, +1 and >+6 charge were excluded from fragmentation, the peptide match feature was set to preferred, the exclude isotope feature was enabled, and selected precursors were dynamically excluded from repeated sampling for 30 s.

Raw data were processed using the MaxQuant software package (version 1.6.0.16[73] and the reference proteome (UniProt Mus musculus Reference Proteome UP000000589, version 2018-07, downloaded 03-08-2018) as well as a database of most common contaminants. The search was performed with full trypsin specificity and a maximum of two missed cleavages at a protein, and peptide spectrum match false discovery rate of 1%. Carbamidomethylation of cysteine residues was set as fixed, oxidation of methionine, and N-terminal acetylation as variable modifications. For label-free quantification, the "match between runs" feature and the LFQ function were activated[74]—all other parameters were left at default.

MaxQuant search results were further processed using the Perseus software package (version 1.6.2.1)[75]. Contaminants, reverse hits, and proteins identified only by site were removed and the log$_2$-transformed LFQ values were used for protein quantification. Mean LFQ intensities of biological replicate samples were calculated, and proteins were filtered for at least two quantified values being present in the three biological replicates. Missing values were replaced with values randomly selected from a normal distribution (with a width of 0.3 and a median downshift of 1.8 standard deviations of the sample population). To determine differentially enriched proteins, we used the LIMMA package[76] in R (version 3.5.1; https://www.r-project.org/). To focus the analysis on interaction partners changing after interferon induction, we filtered for proteins, which were at least twofold enriched above background in at least one condition at an adjusted $p$-value of <0.01, and which showed at least a twofold increase in intensity after interferon induction when compared with steady-state conditions. For this filtered set of proteins, we computed the mean log$_2$ LFQ protein ratio of the interferon-induced (2 and 18 h) and the steady-state condition and used these values to generate a hierarchical cluster analysis and heat maps in Perseus with default settings.

**Targeted mass spectrometry data acquisition and processing**. Parallel-reaction monitoring (PRM) assays were generated based on the shotgun measurements, selecting up to ten high- intensity proteotypic peptides for STAT1, STAT2, STAT3, and IRF9, with no missed cleavages, no methionine, and an even distribution over the chromatographic gradient. PRM assay generation was performed using Skyline[77]. After a test run with a pooled sample showing high target protein expression, we reduced the targets to at least five peptides with a single charge state per protein by further optimizing for best signal-to-noise and an even distribution over the gradient, resulting in a scheduled PRM assay with 4-min windows. Samples were spiked with 100 fmol Pierce Peptide Retention Time Calibration Mixture (PRTC, Thermo-Fisher) to monitor the chromatographic and nanospray stability across the PRM measurements of all samples. For PRM data acquisition, we operated the same instrument type as for shotgun MS, applying a 60-min gradient for separation and with the following MS parameters: survey scan with 60k resolution, AGC 1E6, 50-ms IT, over a range of 400–1300 $m/z$, PRM scan with 30k resolution, AGC 1E5, 300-ms IT, isolation window of 0.7 $m/z$ with 0.2 $m/z$ offset, and NCE of 27%.

Data analysis, manual validation of all peptides and their transitions (based on retention time, relative ion intensities, and mass accuracy), and relative quantification was performed in Skyline (version 4.2). The most intense non-interfering transition(s) of the top five peptides per protein were selected and their peak areas were summed up for peptide quantification (total peak area). Missing peptide intensities were imputed by random values derived from a normal distribution (downshift: median −2.15* standard dev., width: standard dev. *0.05). To correct for minor varying sample injection amounts and instrument stability over the measurements, MS1 signals of six stable background proteins were extracted, selected based on an ANOVA analysis ($q$-value > 0.5, log$_2$ intensity > 25, and standard dev. < 0.2 of MaxQuant LFQ intensities of the shotgun measurements), and used to calculate normalization factors for the PRM data set. After normalization, peptide intensity means were calculated for protein quantification. To ascertain significant interactions, mean log$_2$ protein intensity ratios, standard deviation, and t-test statistics were calculated for each of the target proteins.

**ChIP and ChIP-seq**. In total, $1.5 \times 10^7$ bone marrow-derived macrophages or PMA-treated THP-1 were seeded on a 15-cm dish. The next day, cells were stimulated for 1.5 h either with IFN-β or with IFN-γ. Cells were cross-linked for 10 min at room temperature in 1% formaldehyde PBS (Thermo Fischer, Catalog # 28906). Cells were quenched with 0.125 M glycine for 10 min at RT. Cells were harvested and washed twice with ice-cold PBS. Cells were centrifuged for 5 min at 1350 × $g$ at 4 °C. Cell lysis for BMDM/THP-1 and MEF differed as described: for BMDM and THP-1 cells: pellets were snap frozen in liquid nitrogen and stored at

80 °C overnight. Frozen pellets were thawn on ice for 60 min. Pellets were resuspended in 5 mL of LB1 (50 mM Hepes, 140 mM NaCl, 1 mM EDTA, 10% glycerol, 0.5% NP-40, and 0.25% Triton X-100) by pipetting and rotated at 4 °C for 10 min. Samples were centrifuged for 5 min at 1350 × $g$ at 4 °C. Pellets were resuspended in 5 mL of LB2 (10 mM Tris, 200 mM NaCl, 1 mM EDTA, and 0.5 mM EGTA) by pipetting and rotated at 4 °C for 10 min. Samples were centrifuged for 5 min at 1350 × $g$ at 4 °C. Pellets were resuspended in 3 mL of LB3 (10 mM Tris, 100 mM NaCl, 1 mM EDTA, 0.5 mM EGTA, 0.1% deoxycholate, and 0.5% N-laur-oylsarcosine). Samples were split into 2 × 1.5 mL in 15-mL polypropylene tubes suitable for the Bioruptor® Pico (Diagenode). BioRuptor Sonicator settings: power = high, "on" interval = 30 s, and "off" interval = 45 s, six cycles. Sonicated samples were centrifuged for 10 min at 16000 × $g$ at 4 °C to pellet cellular debris. Chromatin concentration was measured by NanoDrop and 25 μg of chromatin were used for each IP. Three hundred microliters of 10% Triton X-100 were added to each 3 mL of sonicated lysate. Twenty five micrograms of chromatin were stored at 4 °C, which served later on as an input control. The antibody of interest was added to a sonicated chromatin aliquot and mixed (for BMDM: anti-STAT1 Santa Cruz Catalog # sc-346, 2 μl; anti-STAT2 Santa Cruz Catalog # sc-950, 2 μl; IRF9 6F1-H5 supernatant, 150 μl) (for THP-1: anti-STAT1 Cell Signaling Catalog #14995, 10 μl; anti-STAT2 Cell Signaling Catalog # 72604, 10 μl; anti-IRF9 Cell Signaling Catalog # 76685, 10 μl). All samples were filled up to 1 ml with dilution buffer (16.5 mM Tris, pH 8, 165 mM NaCl, 1.2 mM EDTA, 1% Triton X-100, 0.1% SDS, 0.1 mM PMSF, and complete EDTA-free protease inhibitor cocktail (Sigma-Aldrich)). Samples were rotated at 4 °C overnight.

For MEF: The cell pellet was washed with wash buffer I (10 mM HEPES, 10 mM EDTA, 0.5 mM EGTA, and 0.25% Triton X-100) and centrifuged at 4 °C at 450×$g$. The cell pellet was washed with wash buffer II (10 mM HEPES, 200 mM NaCl, 1 mM EDTA, and 0.5 mM EGTA) and centrifuged at 4 °C at 450 × $g$. The cell pellet was resuspended in 800 μl of SDS Lysis buffer (50 mM Tris, pH 8, 10 mM EDTA, 1% SDS, 0.1 mM PMSF, and complete EDTA-free protease inhibitor cocktail (Sigma-Aldrich)) and incubated on a rotating wheel for 1 h at 4 °C.

Lysates were sonicated with the Bioruptor® Pico (Diagenode). BioRuptor Sonicator settings: power = high, "on" interval = 15 s, and "off" interval = 45 s, three cycles. Lysates were centrifuged for 2 × 15 min at 8 °C at maximum speed, and the supernatant was transferred to a new tube. The antibody of interest was added to 25 μg of sonicated chromatin aliquot and mixed (anti-STAT1 Santa Cruz Catalog # sc-346, 2 μl; anti-STAT2 Santa Cruz Catalog # sc-950, 2 μl; IRF9 6F1-H5, 150-μl culture supernatant).

Fifty microliters of magnetic beads (Dynabeads protein G, Life Technologies, 10003D) per sample were blocked overnight in dilution buffer containing 1% BSA at 4 °C. The next day, 50 μl of the beads were added to each sample and incubated at 4 °C while rotating. Afterward, the beads were washed with 1 ml of RIPA buffer (50 mM Tris-HCl, pH 8, 150 mM NaCl, 1% NP-40, 0.1% SDS, 0.5% sodium deoxycholate, and 1 mM DTT), 2× high-salt buffer (50 mM Tris, pH 8, 500 mM NaCl, 0.1% SDS, 1× mM Tris, pH 8, 250 mM LiCl, 0.5% sodium deoxycholate, and 1% NP-40), and TE buffer (10 mM Tris, pH 8, 1 mM EDTA) for 10 min at 4 °C. The samples were eluted in freshly prepared elution buffer (2% SDS, 100 mM NaHCO$_3$, and 10 mM DTT). The cross-link between proteins and DNA was reversed by adding 200 mM NaCl to each sample and incubation at 65 °C at 300 rpm on a rotary shaker for 12 h. Proteinase K, 40 mM Tris, pH 8, and 10 mM EDTA were added to each sample and incubated for 1 h at 55 °C and 850 rpm on a rotary shaker. Each sample was transferred to a phase-lock tube (5Prime), mixed 1:1 with phenol–chloroform–isoamylalcohol (PCI), and centrifuged for 5 min at 12000×$g$. The supernatant was transferred and mixed with 800 μl of 96% ethanol, 40 μl of 3 M CH$_3$COONa, pH 5.3, and 1 μl of glycogen and stored for at overnight at −20 °C. Samples were centrifuged for 45 min at 4 °C and 16,000 × $g$. Pellets were washed in ice-cold 70% ethanol and dried at 65 °C, before diluting the DNA in H$_2$O.

For library generation, the NEBNext Ultra II DNA Library Prep Kit for Illumina from NEB (Catalog # E7645S) was used according to the manufacturer's protocol. The samples were quality checked and sequenced at the Vienna Biocenter Core Facilities NGS Unit.

**ChIP-seq analysis**. For ChiP-seq analysis of BMDM, raw reads were processed using the AQUAS TF pipeline (https://github.com/kundajelab/ChIP-Seq_pipeline; based off Encode phase-3;-idr_thresh 0.01), including alignment against the Mus musculus mm10 genome using BWA (v0.7.13[64]), deduplication using Picard MarkDuplicates (v1.126), Peak Calling using macs2 (v2.1.1[78]), and spp (v1.13[79]). Aligned bam files from MEF and THP-1 ChIP-seq data (Mus musculus genome mm10 or human genome hg38) were filtered to remove reads with low mapping quality (samtools; v1.4; -q 10), overlap with ENCODE blacklisted regions (bedtools; v.25.0; intersectBed) and duplicates (picard; v2.1.1; MarkDuplicates), and used to generate bigwig files (deeptools; v3.1.1; bamCoverage;–normalizeUsing RPGC–binSize 10–extendReads 200).

**Re-ChIP**. The cell lysis and DNA–protein cross-linking was performed as described in the ChIP-seq section. Twenty five micrograms of chromatin were used for each ChIP. Fifty microliters of magnetic Dynabeads (Dynabeads protein G, Life Technologies, 10003D) for each IP were washed twice with PBS. One hundred and fifty microliters of IRF9 antibody (culture supernatant) were added to 850 μl of PBS and

mixed. Beads were added to the antibody and incubated for 2 h at 4 °C. Beads were washed three times with 1 ml at room temperature with 200 mM boric acid, pH 9. In order to cross-link the antibody to the beads, 5.2 mg of dimethylpimelimidate was dissolved in 1 ml of boric acid and added to the magnetic beads and incubated at room temperature on a rotating wheel for 30 min. Beads were washed 5 min each as follows: 2× with Tris 250 mM, pH 8, 2× with PBS, 2× glycine 100 mM, pH 2.5, and 3x with dilution buffer (16.5 mM Tris, pH 8, 165 mM NaCl, 1.2 mM EDTA, 1% Triton X-100, and 0.1% SDS). Fifty microliters of IRF9 cross-linked beads were added to the chromatin samples and incubated overnight on the rotation wheel at 4 °C. The next day, the beads were washed with 1 ml of RIPA buffer (50 mM Tris-HCl, pH 8, 150 mM NaCl, 1% NP-40, 0.1% SDS, 0.5% sodium deoxycholate, and 1 mM DTT), 2× high-salt buffer (50 mM Tris, pH 8, 500 mM NaCl, 0.1% SDS, and 1% NP-40), 2× LiCl buffer (50 mM Tris, pH 8, 250 mM LiCl, 0.5% sodium deoxycholate, and 1% NP-40), and TE buffer (10 mM Tris, pH 8, 1 mM EDTA) for 10 min at 4 °C. Overall, 10 mM DTT was added to the beads and incubated for 30 min at 37 °C and 1400 rpm on a rotary shaker to elute the precipitates. STAT1 (Cell Signaling, Catalog # 14995, 10 μl) and STAT2 (Cell Signaling, Catalog # 72604, 10 μl) antibodies were added to the eluate and filled up 1 ml with dilution buffer (16.5 mM Tris, pH 8, 165 mM NaCl, 1.2 mM EDTA, 1% Triton X-100, 0.1% SDS, 0.1 mM PMSF, and complete EDTA-free protease inhibitor cocktail (Sigma-Aldrich)). Re-ChIP samples were incubated overnight at 4 °C on the rotation wheel. On the next day, samples were washed and eluted as described in the ChIP-seq protocol. ChIP and re-ChIP samples were analyzed by quantitative real-time PCR using the Eppendorf Realplex mastercycler and the SYBR KAPA mix (KAPA SYBR® FAST qPCR Kits, KapaBiosystems). The following PCR program was used: 95 °C for 2 min, 35 × (95 °C for 15 s, 60 °C for 15 s, and 72 °C for 15 s), 95 °C for 15 s, from 60 °C to 95 °C in 20 min for the melting curve, and 60 °C for 2 min. The data were normalized to the input.

**Immunoprecipitation.** In total, $1.5 \times 10^7$ bone marrow-derived macrophages were stimulated for 1.5 h with IFN-β. Cells were lysed in 1 ml of lysis buffer (10 mM Tris-HCl (pH 7.5), 50 mM NaCl, 30 mM NaPPi, 50 mM NaF, 2 mM EDTA, 1% Triton X-100, 1 mM DTT, 0.1 mM PMSF, and 1× protease inhibitor). Cells were incubated for 5 min on ice and then centrifuged for 5 min at 4 °C, $13,400 \times g$. The supernatant was transferred to a new tube. Twenty microliters (10% of the lysate used for the IP) were used as an input control. Two hundred microliters of magnetic beads (Dynabeads protein G, Life Technologies, 10003D) were added to the lysates to preclear for unspecific binding and were rotated for 30 min at room temperature. The precleared lysate was transferred into a new tube.

IgG (Cell Signaling, Catalog #3900S, 1 μl), STAT1 (Cell Signaling, Catalog #14995, 5 μl), and STAT2 (Cell Signaling, Catalog # 72604, 10 μl) antibodies, as well as 80 μl of the IRF9 antibody (culture supernatant) were added to 200 μl of lysate and incubated overnight at 4 °C while rotating. Fifty microliters of magnetic beads were added to each sample and incubated for 3 h at 4 °C while rotating. Afterward, the beads were washed five times with 1 ml of Frackelton buffer and proteins were eluted in SDS sample buffer (250 mM Tris-HCl, pH 6.8, 20% glycerol, 1.6% SDS, 20% β-mercaptoethanol, and 0.002% Bromophenol blue).

**Affinity pulldown of biotinylated ISRE probes.** In total, $1.5 \times 10^7$ Raw 264.7 macrophages were stimulated for 1.5 h with IFN-β. Cells were harvested, washed, and lysed in 1 ml lysis buffer (10 mM Tris-HCl (pH 7.5), 50 mM NaCl, 30 mM NaPPi, 50 mM NaF, 2 mM EDTA, 1% Triton X-100, 1 mM DTT, 0.1 mM PMSF, 1 mM Na₃VO₄, and 1× protease inhibitor). Cells were incubated for 5 min on ice and then centrifuged for 5 min at 4 °C at $13400 \times g$. Two hundred microliters of each sample were incubated with 100 ng of Oas1a (Fw Oas1a oligo 5′ bioteg 5′-TAGAATTTCAGTTTCCATTTCCCGAGAAGGGCA-3′; Rv Oas1a oligo 5′-TGCCCTTCTCGGGAAATGGAAACTGAAAATCTA-3′) and Isgf15 ISRE probes (Fw ISG15 oligo 5′ bioteg 5′- TATTTTCTGTTTCGGTTTCCTTTTCCTAC-3′; Rv ISG15 oligo 5′-GTAGGAAAAGGAAACCGAAACAGAAATA-3′). The reaction was carried out in the presence of 0.1 mM EGTA, 0.5 mM DTT, 40 mM KCl, 1 mM MgCl₂, 500 ng of competitor plasmid, and 2 μl of Poly dI:dC (Thermo-Fisher Scientific, Catalog #20148E). Samples were incubated at room temperature for 30 min while rotating. Fifty microliters of magnetic beads (Pierce Streptavidin beads, Catalog #88817) were added to each sample and the samples were incubated for 10 min at room temperature while rotating. Beads were washed three times with 1 ml of binding buffer and proteins were eluted in SDS sample buffer (250 mM Tris-HCl, pH 6.8, 20% glycerol, 1.6% SDS, 20% β-mercaptoethanol, and 0.002% Bromophenol blue).

**Statistical information.** Q-PCR-derived mRNA expression data as well as ChIP data represent the mean values with standard error of mean (SEM). Differences in mRNA expression data or percent of input data were compared using the one- or two-tailed paired t-test. All statistical analysis was performed using GraphPad Prism (Graphpad) software. Asterisks denote statistical significance as follows: ns, $p > 0.05$; *, $p \leq 0.05$; **, $p \leq 0.01$; ***, $p \leq 0.001$. In all experiments, n in the figure legends represents the number of biological replicates.

Differential expression analysis for BMDM RNA-seq data was carried out using DESeq 2 v1.16.1[69], with an fdr threshold of 0.05. For comparison between BMDM and MEFs, as well as BMDM and THP-1 genes with a minimal lfc ≥ 1 expression and an adjusted p-value ≤ 0.05 were considered differentially expressed.

For ChiP-seq analysis of BMDM, raw reads were processed using the AQUAS TF pipeline (https://github.com/kundajelab/ChIP-Seq_pipeline;based off Encode phase-3;-idr_thresh 0.01), including alignment against the Mus musculus mm10 genome using BWA (v0.7.13[64]), deduplication using Picard MarkDuplicates (v1.126), Peak Calling using macs2 (v2.1.1[78]), and spp (v1.13[79]).

For shotgun proteomics, mean LFQ intensities of biological replicate samples were calculated and proteins were filtered for at least two quantified values being present in the three biological replicates. Missing values were replaced with values randomly selected from a normal distribution (with a width of 0.3 and a median downshift of 1.8 standard deviations of the sample population). To determine differentially enriched proteins, we used the LIMMA package[76] in R (version 3.5.1; https://www.r-project.org/). Hierarchical cluster analysis was performed on standardized (z-scored) data in Perseus with default settings after filtering for proteins, which were at least two-fold enriched above background in at least one condition at an adjusted p-value of < 0.01, and which showed at least a twofold increase in intensity after interferon induction when compared with steady-state conditions.

For targeted proteomics data analysis, manual validation of all peptides and their transitions (based on retention time, relative ion intensities, and mass accuracy), and relative quantification was performed in Skyline (version 4.2). The most intense non-interfering transition(s) of the top five peptides per protein were selected and their peak areas were summed up for peptide quantification (total peak area). Missing peptide intensities were imputed by random values derived from a normal distribution (downshift: median −2.15* standard dev., width: standard dev. *0.05). To correct for minor varying sample injection amounts and instrument stability over the measurements, MS1 signals of six stable background proteins were extracted, selected based on an ANOVA analysis (q-value > 0.5, log₂ intensity > 25, and standard dev. < 0.2 of MaxQuant LFQ intensities of the shotgun measurements), and used to calculate normalization factors for the PRM data set. After normalization, peptide intensity means were calculated for protein quantification. To ascertain significant interactions, mean log₂ protein intensity ratios, standard deviation. and t-test statistics were calculated for each of the target proteins

**Reporting summary.** Further information on research design is available in the Nature Research Reporting Summary linked to this article.

## Data availability

Raw and analyzed data reported in this paper are available under accession number GEO: GSE115435. The mass spectrometry proteomics data have been deposited to the ProteomeXchange Consortium (http://proteomecentral.proteomexchange.org) via the PRIDE partner repository[80] with the data set identifier PXD013209 for the shotgun (interactome) data set or via Panorama Public[81] with the identifier PXD013251 in the case of the targeted MS data. A reporting summary for this article is available as a Supplementary Information File. The source data underlying Figs. 1a, b, 4a–c, 5a, 6b–f, 7a–f and Supplementary Figs. 2a, 3a, 4a, 5a–f, 6a–c, and 7a are provided as a source data file. All other data supporting the findings of this study are available from the corresponding author on reasonable request.

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

## Acknowledgements

Funding was provided by the Austrian Science Fund (FWF) through Grants P 25186-B22, SFB F6103 (to T.D.) and F6101, as well as F6106 (to M.M.). We thank M. Baccarini and G. Versteeg for critical reading of the paper and I. Steinparzer and P. Kovarik for advice concerning the RNA-seq, ChIP-seq, and immunoprecipitation assays. We gratefully acknowledge discussion and suggestions by T. Leonard, D. Levy, and D. Panne that helped us to improve the paper. We also thank E. Ogris, S. Schüchner, and F. Martys for their collaboration in the production of the monoclonal IRF9 antibody. Help and suggestions for ChIP-seq and the establishment of CRISPR–Cas9 knockout cells by A. Vogt, M. Farlik, C. Schmidl, C. Bücker, Stefan Benke, and Gijs Versteeg is gratefully acknowledged. Solexa sequencing was performed by the VBCF NGS Unit (www.vbcf.ac.at).

## Author contributions

E.P. designed and performed experiments, analyzed the results, and contributed to computational analyses of RNA-seq data. D.D., A.S., K.F., and C.C. performed experiments and analyzed the results. M.H. and T.G. performed LC–MS and computational analysis of BioID protein interactors. M.M. provided intellectual input and expertise. M.N. performed computational analyses for RNA-seq and ChIP-seq. T.D. conceived the study, designed experiments, provided intellectual input, helped interpret data, provided supervision, and secured funding. T.D. and E.P. wrote the paper.

## Additional information

**Competing interests:** The authors declare no competing interests.

