## [Peer Review File · Nature Communications]

Reviewers' comments:

Reviewer #1 (Remarks to the Author):

While most of the response to type I IFN has been assigned to a trimer of tyrosine-phosphorylated STAT1, tyrosine-phosphorylated STAT2, and IRF9, there have been scattered reports of “non-canonical” complexes containing a subset of these proteins. In the case of type II IFN, the canonical tyrosine-phosphorylated STAT1 homodimer has been complemented by indications of IRF-based factors and studies of both IFN pathways have produced evidence for non-tyrosine-phosphorylated STAT factors. Furthermore, the status of these proteins with respect to IFN-regulated gene promoters prior to stimulation of cells with IFNs is only beginning to be understood.

This novel and well-designed study takes advantage of multiple omics technologies to add new information and insights to the relatively mature field of gene regulation in the interferon system. The findings demonstrate in a unified system (murine macrophage) the alternative use of IRF9/STAT complexes of various compositions before and after IFN stimulation. Pre-formed IRF9/STAT2 is apparently regulating low basal expression of ISGs in murine macrophages, and IFN stimulation leads to ISGF3 trimers, though evidence suggests that this trimer forms on DNA rather than in solution. Together these new findings may ultimately enable new insights into both antiviral immunity and IFN-based autoimmune diseases.

Despite the overall enthusiasm for this study, one big question is whether similar factors are used in (1) different mouse cell types, and (2) in any human cell types. Due to species-specificity of IFNs, IFN signaling, and STAT proteins, the generality of the findings and translation to humans is an important consideration.

In addition, a few questions arise that should be addressed:

1. Line 107 Figure 2 text reference should be corrected. In this figure it is unclear what time points are represented and if any time course data are available. The genome browser panels do not have axis labels, and the relative abundance of peak signals is unknown- does the legend indicate all axes are 0-200? Please clarify or show a combined plot on a single scale.

2. The CHIP data in fig 2 show peaks but not complexes. Please provide some CHIP-ReCHIP data to test if complexes/proteins are on the same genes at the same time.

3. The western blots are unfortunately the weakest data in the paper. Panels are too close-cropped around bands, the backgrounds in some samples are high enough to defy interpretation of “more or less” (for example is there more STAT2 after IFN β ?), and questions about antibody sensitivity make comparisons to CHIP-PCR difficult (especially the very weak STAT1 signals). Loading controls and quantification are absent. See my comment #2 for more clarity via Chip-reChip.

4. The data on IRF9 nuclear localization in Fig. 5a are not unexpected based on the prior definition of the IRF9 NLS and its ability to be regulated by STAT2 (<https://doi.org/10.1073/pnas.97.13.7278>). IRF9 is in nucleocytoplasmic flux and defaults to the nucleus and is redistributed by STAT2 presence or absence. It is difficult to know from data presented if what is detected in the nucleus is stable or you are freeze-framing the moving proteins. This should be demonstrated or else modify the breadth of interpretations.

5. The experiments in fig. 6A do not have an IFN-stimulated control by which to demonstrate the effect and efficacy of JAK inhibition in the same experiment. These data are essential to interpretation.

6. The results of Fig. 6C provide an exceptional insight that in deficiency the ISGF3 proteins appear to all regulate one another. However there is no genetic complementation to prove the point. Expression of the absent protein should be shown to revert the phenotype.

7. Figure 7- not clear what the arrows emanating from the nucleus signify in Homeostasis panel.

8. A few comments about the discussion. It is a very well written discussion that reflects deep thinking by the authors, but some questions and curiosities arose in reading it.

-Line 247- may be a typo or just unclear on the meaning, please define the “transcriptional mode switch”

-Line 269- Is there truly evidence of complexes exchanging? Seems it could be accomplished by addition of STAT1.

-Line 275- agree that IRF9-STAT2 fusion protein supports the general concept, but this paper also reported IRF9-VP16 could do the same thing. This makes me wonder if there is anything special about the STAT2 TAD at all, or is it merely an IFN-sensor relay system.

-Line 299- Seems that the biotinylation of proteins involved in chromatin organization would occur for many BirA-tagged DNA-binding proteins, and this does not necessarily need to invoke any role of IRF9 in actively altering the landscape. IRF9 may bind to nucleosome free regions or those with specific nucleosome modifications. You have not provided data on nucleosome presence on these promoters, though there is some available (<https://doi.org/10.1016/j.cell.2015.12.032>) that you might compare at key loci. Similarly for the discussion of not finding the essential RVB1/2- specificity of the interaction identified in BirA fusions – as well as time course and abundance of myriad protein factors come into question. STAT2 is quite different in mouse and human. Does mouse even require RVBs?

9.- Minor point- the bibliography needs editing for uniform stylistic issues.

Reviewer #2 (Remarks to the Author):

The manuscript by Platanitis et al. examines molecular complexes that regulate IFN-induced and tonic signaling to ISGs in mouse macrophages. They address whether the full ISGF3 complex mediates tonic signaling and expression of ISGs, and demonstrate that a constitutive Stat2-Irf9 complex (and not ISGF3) mediates expression of ISGs in the resting state. The manuscript is well written and clearly presented, and the data supports the claims. Importantly, this work clarifies the independent roles of Stat1 and Stat2-Irf9 complexes in the resting state with careful transcription and proteomic assays. This paper should be of interest to researchers interested in signaling mechanisms in macrophages and specifically with regards to the cells anti-viral response. I have only minor comments for the authors.

Minor comments.

1. I would be informative to see a motif analysis that recapitulates the known motifs from the various CHIP bound categories. Specifically, how do the motifs compare between the Stat2-IRF9-bound and ISGF3-bound promoters? And are there differences in the DNA sequence for sites that

remain only Stat2-Irf9 bound versus those that become full ISGF3 sites upon stimulation. This could be accomplished readily with de novo motif finding on the bound promoters.

2. Line 107: "(Fig. 2a, d." should be "(Fig. 2a, d)."

3. Figure 4. Can the authors comment on the lack of overlap between what is seen with Irf9 BioID and the Stat2 BioID. In other words, do the authors propose IRF9 interactions and complexes that are independent of Stat2 that would explain the much larger number of IFN-gamma-specific Irf9 components?

4. Figure 5. The small bars beside Stat2 are confusing – what are they meant to indicate? How is the small bar to the left of the text related to Stat2 versus the larger bar underneath?

5. While it is beyond the scope of this work to perform all experiments in human cells, assuming suitable reagents are available, it would be helpful if the authors could demonstrate (if even only by the western of cell fractions shown in Figure 5b) that similar lack of Stat1 in the resting state is also seen in human cells.

Reviewer #3 (Remarks to the Author):

In the manuscript "Homeostatic and Interferon-induced gene expression represent different states of 3 promoter-associated transcription factor ISGF3" Platanitis et al investigate the interactions, localization and activity of the components of the ISGF3 complex, STAT1, STAT2 and IRF9 under homeostatic and interferon-induced conditions. Using a combination of ChIP-seq, transcriptome as well as protein-protein interaction data the authors come up with a new model in which under homeostatic conditions the majority of IRF9 is localized in the nucleus and forms a complex with STAT2 to regulate basal expression of ISGs, which they explain as a mechanism to prime the cells for enhanced cytokine responsiveness. Under IFN-induced conditions the STAT2-IRF9 complexes are replaced for ISGF3 complexes to accommodate for rapid IFN-induced expression.

Based on my expertise in quantitative proteomics and proximity labeling I am mainly commenting on this aspect of the manuscript. The authors claim in their final model that STAT2 and IRF9 form a complex localized in the nucleus under steady state condition and that association with STAT1 to form the ISGF3 complex is enhanced under IFN-induced conditions. Unfortunately, none of the data that the authors provide on protein-protein interactions, including affinity purification followed by WB and BioID followed by mass spectrometry, do not show an enhanced association of STAT1 with STAT2/IRF9 under IFN-induced conditions. In figure 3a the STAT1 IP hardly purified any STAT2 and no IRF9 in homeostatic as well as IFN-induced conditions. Additionally, in the BioID experiment for IRF9 the authors do not detect any STAT1 neither in basal nor in IFN-induced conditions. While STAT1 is identified in the BioID experiment of STAT2, there is no difference visible between unstimulated and IFN-induced conditions. Therefore, currently the experiments presented do only support the model in which STAT2 interacts with IRF9 under basal conditions, but there is no evidence provided for the dynamics described for IFN-induction.

Some more detailed comments regarding the BioID experimental setup, analysis and presentation of the data:

- Selection of controls:

For the BioID experiment the authors chose a myc-tagged BirA construct as control, which I think might not be a suitable control. The background of proximity biotinylation experiments is very much dependent on the localization of the bait protein. Therefore, in order to determine the interactors of a bait, it is crucial to select a control that localizes in the same cellular compartment than the bait itself. The authors show that most of IRF9 at least is localized in the nucleus in all conditions and to determine specific interactors of IRF9, it would be important to have a control that also localizes in the nucleus. Otherwise, there is a high chance of claiming nuclear proteins as specific interactors that are only higher abundant in the bait-BioID sample because of the different localization of the control and the bait protein. The authors should repeat the experiment for IRF9 including a nuclear localized control or demonstrating that myc-BirA is in fact localized in the nucleus.

- Differential interactome under IFN-induced conditions:

For the BioID experiment the expression of the BirA Fusion constructs is induced, Biotin is added for 18h and IFN induction is performed for 1.5h. I am assuming that the IFN induction is performed at the end of the 18h biotin addition. My concern regarding this experimental setup is that most of the biotin labeling is performed under basal conditions and only for a much shorter time the labeling happens under IFN-induced condition. I am wondering if this is one of the reasons why the authors do not see much effects on the interactomes comparing basal and IFN-induced conditions.

- Replication:

Especially for determining quantitative differences of interactors under various conditions the authors should include at least 3 replicates of each condition for a better quantitative comparison and apply more rigorous statistics. Replication would also help to confidently quantify smaller abundance changes across conditions.

- PRM analysis:

The authors perform a targeted PRM experiment monitoring STAT1, STAT2 and IRF9 in BioID experiments under basal conditions in two replicates. Why were the IFN-induced conditions not included? This targeted experiment could potentially show better quantitative differences in complex formations between the three components under basal and stimulated conditions. Figure 4a and 4e are confusing since they plot 2 replicates next to each other. It would be much better if the authors run the experiment in triplicates and summarize the data as mean ratios with standard deviations as well as include the IFN-induced conditions.

- STAT1 BioID experiments:

Why did the authors do not perform BioID experiments on STAT1 for PRM and shotgun proteomic experiments?

- Figure 4 comments:

Figure 4 c and 4d visualize Gene Ontology enrichment for the IRF9 interactors. Several things are not clear from the figure representation and the figure description. In the current form these figure panels are not very informative.

1. As mentioned above the claimed interactors might be based on a control that does not co-localize with IRF9, so I would assume that the interactome contains unspecific interactors.
2. Why are some of the nodes labeled with bold font while other aren't?
3. It is not clear what the different colors represent?
4. Why are some of the nodes bigger than others?
5. Why do the authors refer in the figure description to Supplementary Figures 1, 3 and 4?

- Public availability of RAW data:

All acquired RAW data, PRM as well as shotgun proteomic data should be made publicly available for example through PRIDE ProteomeExchange and Panorama (for Skyline data)

Some general comments:

- References to published literature are missing in several places (Page 2 line 46; Page 3 line 50; Page 3 line 70; Page 5 line 126)
- Some of the WBs in Figure 3 are not very conclusive, especially the WBs for STAT2 IPs. The authors claim that more STAT2 is associated with IRF9, however based on the STAT2 band, it appears that more STAT2 is purified which could also lead to an increase in IRF9. These WBs should be

repeated. Since the differences look rather small, it would be important to replicate these experiments and perform a quantification of the WB signal.

- Figure 7: A more detailed description of the model in the figure description is missing including the meaning of the arrows.

REPLY TO THE REFEREES' COMMENTS

Reply to all referees:

The paper has been extensively modified with new data. The most substantial revisions are two new RNA-seq and ChIP-seq datasets in new figures 3 and Supplementary Fig.4 and a complete rerun of the BioID/PRM data in triplicates, including an analysis of Stat1-BirA* cells and normalization of all BioID data to mycBirA* or mycNLS-BirA* controls (new figures 4,5 and Supplementary Fig.5 and Supplementary Fig.6).

In addition, the following figures contain new material:

- ATACseq tracks were added to figures 2A and Supplementary Fig.2b.
- New figure 7e shows STAT/IRF9 expression in reconstituted Stat1 ko MEFs.
- ChIP-reChIP analysis has been performed and is shown in new Supplementary figure 3a
- A STAT2-IRF9/ISGF3 binding site analysis is shown in the new Supplementary Fig.3b, c.
- The impact of the Jak inhibitor on IFN-induced ISG expression is shown in new Supplementary Fig.7.
- To reduce the length of the manuscript two figures were deleted/modified. The original figure 5b was excluded from the manuscript because we decided to show all inhibition studies in bone marrow macrophages and use only the pathway- specific JAK inhibitor. Thus, the figure showing Staurosporine effects in Raw macrophages has become obsolete. Original figure S3A was omitted because figure 5a clearly demonstrates the IFN response of Raw macrophages and we therefore deemed a blot showing the IFN responsiveness of the cells to contain redundant information. Supplementary Fig.5 no longer shows the entire Dox titration curve of the cells containing BirA* fusion genes, but just the concentrations used in our experiments.
- Following advice from a native speaker we changed the title to clearer reflect the main message of the paper.
- The mass spectrometry data have been deposited at the Proteome Xchange Consortium and can be accessed under USERNAME: reviewer29395@ebi.ac.uk; PASSWORD: 2a7hQvvQ.
- PRM data have been submitted to the panorama public server, but we have not yet received an identifier. We will submit this to Nature Comm. once we have received it.

Together the new data have strengthened the main conclusions of our paper. We thank all referees for their constructive suggestions and hope they find this improved version suitable for publication.

Reply to the comments of individual reviewers:

Reviewer #1 (Remarks to the Author):

Despite the overall enthusiasm for this study, one big question is whether similar factors are used in (1) different mouse cell types, and (2) in any human cell types. Due to species-specificity of IFNs, IFN signaling, and STAT proteins, the generality of the findings and translation to humans is an important consideration.

We address this important question with a considerable amount of new data, specifically RNAseq and ChIPseq data from mouse fibroblasts and human THP1 cells, both wt and IRF9-deficient (new figs 3 and Supplementary Fig.4). In brief, the fibroblast data support the findings in macrophages. The THP1 data are ambiguous. While there is a clear reduction of basal ISG expression and STAT1/STAT2 protein levels drop in IRF9-deficient cells, we cannot clearly assign ChIPseq peaks to ISGs in untreated cells. Similarly, a dataset from ref 49 in HeLa cells does not support constitutive STAT2/IRF9 binding to chromatin. Unfortunately, there is no other ChIPseq dataset addressing STAT2 binding in human cell

types in public databases. ChIP-ChIP data in ref 50 rather support our model, clearly stating binding of STAT2 to ISG promoters in untreated human hepatocytes. Thus, the final verdict on the applicability of our findings in mouse macrophages to human cells needs yet to be announced. We discuss the situation on p. 13 line 338 to p. 14, line 350.

In addition, a few questions arise that should be addressed:

1. Line 107 Figure 2 text reference should be corrected. In this figure it is unclear what time points are represented and if any time course data are available. The genome browser panels do not have axis labels, and the relative abundance of peak signals is unknown- does the legend indicate all axes are 0-200? Please clarify or show a combined plot on a single scale.

The missing information (treatment duration and scales in the ChIPseq tracks) are now indicated in the figure legend. We have chosen to look at an early timepoint of treatment (1.5 h) to rule out secondary effects of the delayed response. The scales are now 0-150 for all tracks).

2. The ChIP data in fig 2 show peaks but not complexes. Please provide some ChIP-ReChIP data to test if complexes/proteins are on the same genes at the same time.

We provide a ChIP-reChIP experiment in the new supplementary Fig. 3a. The subunits in IFN-treated cells are beyond doubt combined on one promoter after IFN treatment. In untreated cells the assay is very close to the detection limit but the data we manage to extract support STAT2-IRF9 on the same promoters.

3. The western blots are unfortunately the weakest data in the paper. Panels are too close-cropped around bands, the backgrounds in some samples are high enough to defy interpretation of "more or less" (for example is there more STAT2 after IFN β ?), and questions about antibody sensitivity make comparisons to ChIP-PCR difficult (especially the very weak STAT1 signals). Loading controls and quantification are absent. See my comment #2 for more clarity via Chip-reChip.

We provide new and improved western blots for most experiments (figures 4a; 6b,c,d). All blots contain loading controls and where necessary have been quantified (figures 4b, 6e,). Please note that some signals, e.g. the colIP of STAT1 and STAT2 from untreated cells are weak and remain weak under many tested conditions This includes using antibodies from different commercial and home-made sources. The result appears to reflect an intrinsic property of this complex and is entirely consistent with other attempts (refs 22-24) to identify STAT2/STAT1 interaction in absence of overexpression. A second important point is that the crops of our western blots are not done in silico, but rather result from the need to cut the membrane to separate STAT1 and STAT2 for antibody staining of the same blot. Stripping and reprobing reduces the signal-to-noise ratio and does not work well in our hands, particularly when examining colIPs.

4. The data on IRF9 nuclear localization in Fig. 5a are not unexpected based on the prior definition of the IRF9 NLS and its ability to be regulated by STAT2 (<https://doi.org/10.1073/pnas.97.13.7278>). IRF9 is in nucleocytoplasmic flux and defaults to the nucleus and is redistributed by STAT2 presence or absence. It is difficult to know from data presented if what is detected in the nucleus is stable or you are freeze-framing the moving proteins. This should be demonstrated or else modify the breadth of interpretations.

We are aware of the important data from the Reich and Horvath labs showing regulation of STAT2 and IRF9 localization (refs 39, 40). We agree that IF data are a snapshot of a dynamic situation and we do not intend to say that they 'prove' a nuclear function of IRF9 (see p. 10, lines 252-255). Our data show that under wt conditions only part of IRF9 is associated with STAT2, leaving the rest to shuttle and localize to the nucleus. This explains why IF data show a strong nuclear signal with our IRF9 antibody. The STAT2-associated fraction is of most interest to us because it is associated with ISG promoters in resting cells. To reconcile the export signal provided by STAT2 with resting-state DNA association of STAT2/IRF9 we assume that part of the complexes are trapped on chromatin and temporarily excluded from shuttling. We modified the discussion on p 14, line 364 to account

for our interpretation of the data.

5. The experiments in fig. 6A do not have an IFN-stimulated control by which to demonstrate the effect and efficacy of JAK inhibition in the same experiment. These data are essential to interpretation.

The efficacy of inhibition was shown in fig. 6e by an analysis of STAT1 and STAT2 tyrosine phosphorylation. The requested control showing the degree at which gene expression is inhibited is now included as well (Supplementary Fig.7).

6. The results of Fig. 6C provide an exceptional insight that in deficiency the ISGF3 proteins appear to all regulate one another. However there is no genetic complementation to prove the point. Expression of the absent protein should be shown to revert the phenotype.

We now show that complementation of STAT1-deficient cells indeed restores wt levels of STAT2 and IRF9 expression (new Fig. 7e). We were not able to sort, regrow and test cultures from single clones with inducible STAT2 or IRF9 without requiring substantially more time than the already extended period of revision.

7. Figure 7- not clear what the arrows emanating from the nucleus signify in Homeostasis panel.

We revised the figure to keep it more simple and reduce the message to the key points. We hope our attempts are successful.

8. A few comments about the discussion. It is a very well written discussion that reflects deep thinking by the authors, but some questions and curiosities arose in reading it.

We thank the referee for insightful comments on the discussion that had us revisit some of our concepts.

-Line 247- may be a typo or just unclear on the meaning, please define the "transcriptional mode switch"

Thank you for pointing this out. We changed the wording to 'molecular switch' which we hope is clearer.

-Line 269- Is there truly evidence of complexes exchanging? Seems it could be accomplished by addition of STAT1.

In fact, we would have liked to propose STAT1 addition as a more elegant mechanism. In the end we did not dare based on the current wisdom that particularly nuclear STAT1 is all but dimeric and that data from the Darnell and Panne labs suggest that ISGF3 contains one copy of each STAT1, STAT2 and IRF9. Under the circumstances we don't see alternatives to proposing exchange based on a most likely higher off-rate of the STAT2/IRF9 dimer compared to ISGF3 where STAT1 reportedly makes additional DNA contacts (ref 53). Our data would best be reconciled with data in the literature by proposing that ISGF3 forms on DNA from a fully phosphorylated STAT1-STAT2 heterodimer and (loosely preassociated?) IRF9 (revised discussion, p. 365-p. 15, line 373).

-Line 275- agree that IRF9-STAT2 fusion protein supports the general concept, but this paper also reported IRF9-VP16 could do the same thing. This makes me wonder if there is anything special about the STAT2 TAD at all, or is it merely an IFN-sensor relay system.

We think it is a sensor relay system, but one that would not work with all transactivation domains (TAD). Available data (e.g. doi: 10.1074/jbc.274.36.25343) and the fact that the STAT2, but not the STAT1 TAD is needed for function of the ISGF3 complex lead us to hypothesize that the STAT2 TAD has a distinct mode of addressing the transcriptional machinery, representing a subclass of TADs that, if tested, would work when fused to IRF9. The specific attributes of this TAD subclass and the difference from others pose an interesting problem for future research.

-Line 299- Seems that the biotinylation of proteins involved in chromatin organization would occur for many BirA-tagged DNA-binding proteins, and this does not necessarily need to invoke any role of IRF9 in actively altering the landscape. IRF9 may bind to nucleosome free regions or those with specific nucleosome modifications. You have not provided data on nucleosome presence on these promoters, though there is some available (<https://doi.org/10.1016/j.cell.2015.12.032>) that you might compare at key loci. Similarly for the discussion of not finding the essential RVB1/2- specificity of the interaction identified in BirA fusions – as well as time course and abundance of myriad protein factors come into question. STAT2 is quite different in mouse and human. Does mouse even require RVBs?

We agree that the impact of STAT2-IRF9 on chromatin structure needs to be addressed in future research and is currently not clear. That being said, we found a recently published ATACseq dataset from resting bone marrow-derived macrophages (ref 18) and are showing exemplary tracks in revised figures 2 and Supplementary Fig. 2b. In most, but not all ISGs the ATACseq signals correspond very well with the localization of the STAT2/IRF9 peaks. Regarding the interactor profile: in response to referee 3 we included STAT1-BirA, added a BirA-NLS control and triplicated the interactor screen with all different cell lines, including the parallel reaction monitoring (PRM). We also included two time points of IFN β treatment and IFN γ -treated cells in our analysis. The PRM data for STAT1, STAT2, IRF9 and, as a control for STAT1 interaction, STAT3, are shown from untreated and IFN-treated cells (revised figs. 4,5, Supplementary Fig 6). This improved and extended PRM data set agrees with our notion that there is a constitutive STAT2/IRF9 complex, but no STAT1/IRF9 interaction before IFN treatment. In contrast, STAT1 shows proximity to STAT2 and STAT3 in resting cells, and, as expected, to STAT2, STAT3 and IRF9 in IFN-treated cells. With regard to the entire interactor screen we have simplified the data representation (fig 5b, supplementary tables 7,8).

RVB 2 was found in our screen in proximity to IRF9 and STAT2 and it passed the padj. cutoff of <0.01 , but it did not quite make the $LFC > 1$ cutoff in the triplicates (see discussion). In contrast, no proximity to STAT1 above background was found. We think this argues for a similar role of RVB2 in mouse and humans but have no data that directly address this.

9.- Minor point- the bibliography needs editing for uniform stylistic issues.

We went through the references and hope to have found all deviations from the required style.

Reviewer #2 (Remarks to the Author):

Minor comments.

1. I would be informative to see a motif analysis that recapitulates the known motifs from the various CHIP bound categories. Specifically, how do the motifs compare between the Stat2-IRF9-bound and ISGF3-bound promoters? And are there differences in the DNA sequence for sites that remain only Stat2-Irf9 bound versus those that become full ISGF3 sites upon stimulation. This could be accomplished readily with de novo motif finding on the bound promoters.

Following the referee's suggestions, we performed a de novo computational analysis of ISRE sequences in genes that switch from STAT2-IRF9 to ISGF3 and those that are

associated with STAT2-IRF9 after IFN treatment (new supplementary Fig. 3b,c). The differences are quite subtle.

2. Line 107: "(Fig.. 2a, d." should be "(Fig. 2a, d)."

The mistake has been corrected, thank you for pointing it out.

3. Figure 4. Can the authors comment on the lack of overlap between what is seen with Irf9 BioID and the Stst2 BioID. In other words, do the authors propose IRF9 interactions and complexes that are independent of Stat2 that would explain the much large number of IFN-gamma-specific Irf9 components?

The absence of a large spectrum of common interactors would best be explained by different STAT-IRF9 complexes determine a different spectrum of interactors. Specifically, we agree with the referee that STAT2-IRF9 and IRF9 either alone or in a different complex bind to different partner proteins. We mention this possibility in the results section (page 9, lines 228-229).

4. Figure 5. The small bars beside Stauro are confusing – what are they meant to indicate? How is the small bar to the left of the text related to Stuario versus the larger bar underneath?

We apologize for this error which-strangely enough-was visible on adobe software but not on the 'preview' app. The figure has anyway become obsolete as pointed out in our reply to all referees.

5. While it is beyond the scope of this work to perform all experiments in human cells, assuming suitable reagents are available, it would be helpful if the authors could demonstrate (if even only by the western of cell fractions shown in Figure 5b) that similar lack of Stat1 in the resting state is also seen in human cells.

In reply also to a comment from referee 1 we address this important question with RNAseq and ChIPseq data from human THP1 cells, both wt and IRF9-deficient, to assess the role of the STAT2/IRF9 complex (new figs 3 and Supplementary Fig.4). In brief, there is a clear reduction of basal ISG expression in IRF9-deficient cells and STAT1/STAT2 protein levels drop. However, we cannot clearly assign ChIPseq peaks to ISGs in untreated cells. Similarly, a dataset from ref 49 in HeLa cells does not support constitutive Stat2/IRF9 binding to chromatin. Unfortunately, there is no ChIPseq dataset showing STAT2 binding in other human cell types in public databases. We think the lack of constitutive STAT2/IRF9 peaks may well be a sensitivity issue because less reads are mapped in both datasets to ISREs of IFN-treated cells compared to mouse macrophages or fibroblasts, despite a larger number of total reads. ChIP-ChIP data in ref 50 rather support our model, clearly stating binding of STAT2 to ISG promoters in untreated human hepatocytes. Thus, the final verdict on the applicability of our findings in mouse macrophages to human cells needs yet to be announced. We discuss the situation on p. 13 line 338 to p. 14, line 350.

Reviewer #3 (Remarks to the Author):

Based on my expertise in quantitative proteomics and proximity labeling I am mainly commenting on this aspect of the manuscript. The authors claim in their final model that STAT2 and IRF9 form a complex localized in the nucleus under steady state condition and that association with STAT1 to form the ISGF3 complex is enhanced under IFN-induced conditions. Unfortunately, none of the data that the authors provide on protein-protein interactions, including affinity purification followed by WB and BioID followed by mass spectrometry, do not show an enhanced association of STAT1 with STAT2/IRF9 under IFN-induced conditions. In figure 3a the STAT1 IP hardly purified any STAT2 and no IRF9 in homeostatic as well as IFN-induced conditions. Additionally, in the BioID experiment for IRF9 the authors do not detect any STAT1 neither in basal nor in IFN-induced conditions. While STAT1 is identified in the BioID experiment of STAT2, there is no difference visible between unstimulated and IFN-induced conditions. Therefore, currently the experiments presented do only support the model in which STAT2 interacts with IRF9 under basal conditions, but there is no evidence provided for the dynamics described for IFN-induction.

We apologize for not being clearer about our interpretations: one of our key points is that contrasting current wisdom-there is NO pre-association of STAT1 and the STAT2/IRF9 complex in the cytoplasm or nucleoplasm. We propose that ISGF3 assembles on DNA mainly after IFN treatment. The relative amount of the DNA-associated ISGF3 in untreated cells is very small. Therefore, a STAT1-IRF9 colIP is below the detection limit. However, STAT1-BirA* causes biotinylation of IRF9 after IFN treatment, strongly suggesting that it reports proximity on DNA. Antibody-mediated colIPs don't precipitate an entire ISGF3 complex because they are performed in absence of DNA. In contrast, DNA-mediated precipitation pulls down a complete ISGF3 after IFN treatment, in line with the BioID approach.

STAT1 and STAT2 form heterodimers which-despite trying various conditions- are barely above detection levels in colIP, but readily detectable by affinity labeling in both STAT1-BirA* and STAT2-BirA* cells. We don't know the reason for the difference between colIP and BioID, but the assumption that STAT1-STAT2 heterodimers are rather unstable appears to be in line also with experiments in the literature showing weak interaction by colIP in absence of overexpression (refs 22-24).

Some more detailed comments regarding the BioID experimental setup, analysis and presentation of the data:

- Selection of controls:

For the BioID experiment the authors chose a myc-tagged BirA construct as control, which I think might not be a suitable control. The background of proximity biotinylation experiments is very much dependent on the localization of the bait protein. Therefore, in order to determine the interactors of a bait, it is crucial to select a control that localizes in the same cellular compartment than the bait itself. The authors show that most of IRF9 at least is localized in the nucleus in all conditions and to determine specific interactors of IRF9, it would be important to have a control that also localizes in the nucleus. Otherwise, there is a high chance of claiming nuclear proteins as specific interactors that are only higher abundant in the bait-BioID sample because of the different localization of the control and the bait protein. The authors should repeat the experiment for IRF9 including a nuclear localized control or demonstrating that myc-BirA is in fact localized in the nucleus.

We now performed proximity labeling with both myc-BirA and myc-NLS-BirA controls (Fig. 4,5). We also show the nuclear/cytoplasmic distribution of both control proteins (Supplementary Fig.6a,b). Based on its distribution, myc-NLS-BirA is the better choice for IRF9 and also for STATs in IFN-treated cells, but myc-BirA better resembles the predominantly cytoplasmic localization of STATs in untreated cells. Accordingly, we normalized STAT1-BirA* and STAT2-BirA* in resting cells to myc-BirA* controls and all other situations to the myc-NLS-BirA* control. While the choice of control influences the overall spectrum of affinity-labelled proteins (see also below) normalization to both controls supports the conclusion about STAT2/IRF9 and STAT1/STAT2 complexes or the absence of proximity between STAT1 and IRF9 in resting cells.

- Differential interactome under IFN-induced conditions:

For the BioID experiment the expression of the BirA Fusion constructs is induced, Biotin is added for 18h and IFN induction is performed for 1.5h. I am assuming that the IFN induction is performed at the end of the 18h biotin addition. My concern regarding this experimental setup is that most of the biotin labeling is performed under basal conditions and only for a much shorter time the labeling happens under IFN-induced condition. I am wondering if this is one of the reasons why the authors do not see much effects on the interactomes comparing basal and IFN-induced conditions.

The interferon-induced change in the interactome is now shown in the new figure 5b). We initially chose the short time point because IFN treatment changes the stoichiometry of the STATs (they are encoded by IFN-induced genes), hence the complexes they form. The early time point also allows for a better integration of ChIP-seq and RNA-seq results. We did however analyze a late time point normalized to an identically treated NLS-BirA* control as suggested (Fig. 5b). The results do not change our conclusions with regard to the proximity of ISGF3 subunits.

- Replication:

Especially for determining quantitative differences of interactors under various conditions the authors should

include at least 3 replicates of each condition for a better quantitative comparison and apply more rigorous statistics. Replication would also help to confidently quantify smaller abundance changes across conditions.

All experiments shown have now been performed in triplicate samples.

- PRM analysis:

The authors perform a targeted PRM experiment monitoring STAT1, STAT2 and IRF9 in BioID experiments under basal conditions in two replicates. Why were the IFN-induced conditions not included? This targeted experiment could potentially show better quantitative differences in complex formations between the three components under basal and stimulated conditions. Figure 4a and 4e are confusing since they plot 2 replicates next to each other. It would be much better if the authors run the experiment in triplicates and summarize the data as mean ratios with standard deviations as well as include the IFN-induced conditions.

This has been done as suggested by the referee (revised figs. 4c and Supplementary Fig. 6c, supplementary table 6).

- STAT1 BioID experiments:

Why did the authors do not perform BioID experiments on STAT1 for PRM and shotgun proteomic experiments?

STAT1 BioID has now been included (revised figs 4,5 and Supplementary Fig 6.)

- Figure 4 comments:

Figure 4 c and 4d visualize Gene Ontology enrichment for the IRF9 interactors. Several things are not clear from the figure representation and the figure description. In the current form these figure panels are not very informative.

We have changed new figure 5b and supplementary table 7 to contain data that can be extracted from our screen without extensive validation, hoping that this is both informative and clear.

1. As mentioned above the claimed interactors might be based on a control that does not co-localize with IRF9, so I would assume that the interactome contains unspecific interactors.

We included a NLS-BirA* control (figures 4c, 5b, supplementary Fig. 6c, supplementary tables 6,7,8) and included the subcellular distribution of both BirA* and NLS-BirA* (figs. S6a, b)

2. Why are some of the nodes labeled with bold font while other aren't?

3. It is not clear what the different colors represent?

4. Why are some of the nodes bigger than others?

2.-4.: The node display has been omitted.

5. Why do the authors refer in the figure description to Supplementary Figures 1, 3 and 4?

The supplementary figures contained information about the experimental set-up, the establishment of the cells expressing BirA* fusion proteins and a heat map of interactors.

With the new display the reference to figure 4 has become obsolete and the entire sentence has been removed because it caused confusion more than being helpful.

- Public availability of RAW data:

All acquired RAW data, PRM as well as shotgun proteomic data should be made publicly available for example through PRIDE ProteomeExchange and Panorama (for Skyline data)

We now show triplicate datasets from PRM data probing the proximity between STAT1, STAT2 and IRF9. As a control we included STAT3 which forms heterodimers with STAT1, to provide additional evidence that our STAT1-BirA cells reproduce published data. PRM data have been uploaded to the PRIDE Proteome Exchange and Panorama (for Skyline data) database.

Some general comments:

- References to published literature are missing in several places (Page 2 line 46; Page 3 line 50; Page 3 line 70; Page 5 line 126)

All references have been inserted as suggested.

- Some of the WBs in Figure 3 are not very conclusive, especially the WBs for STAT2 IPs. The authors claim that more STAT2 is associated with IRF9, however based on the STAT2 band, it appears that more STAT2 is purified which could also lead to an increase in IRF9. These WBs should be repeated. Since the differences look rather small, it would be important to replicate these experiments and perform a quantification of the WB signal.

We have repeated and quantified the western blot as suggested (revised figure 4,6,7).

- Figure 7: A more detailed description of the model in the figure description is missing including the meaning of the arrows.

We redesigned this figure and its legend with (hopefully) improved clarity.

REVIEWERS' COMMENTS:

Reviewer #1 (Remarks to the Author):

The revised manuscript has been greatly improved by addressing this referee's comments with additional experimental data, greater clarity in writing, and careful consideration of conclusions and interpretations. Although every comment was not able to be addressed directly the paper is a tour de force and provides great insight into the interferon response system at baseline and following stimulation.

Reviewer #2 (Remarks to the Author):

I am satisfied that the revised manuscript has addressed all my previous concerns

Reviewer #3 (Remarks to the Author):

In the revised version of the manuscript the authors have addressed all my comments sufficiently and I have no further request in providing more data.

However, I would like to see an improved visualization of some of the new datasets in Figure 5b.

The z-score visualization seems misleading, The first column of the heatmap has mostly negative values (z-scores). A reader could think that these ones have a negative log₂ fold change compared to the BirA*.

In my opinion it would be better to use the respective controls as described here to determine the specific interactors of STAT1, STAT2 and IRF9. Once this list of proteins is determined, it would be good to look how these proteins are changing over time and provide fold changes comparing the uninduced protein of interest with the IFN stimulated conditions.

Reply to referee #3:

Reviewer #3 (Remarks to the Author):

In the revised version of the manuscript the authors have addressed all my comments sufficiently and I have no further request in providing more data.

However, I would like to see an improved visualization of some of the new datasets in Figure 5b.

The z-score visualization seems misleading. The first column of the heatmap has mostly negative values (z-scores). A reader could think that these ones have a negative log₂ fold change compared to the BirA. In my opinion it would be better to use the respective controls as described here to determine the specific interactors of STAT1, STAT2 and IRF9. Once this list of proteins is determined, it would be good to look how these proteins are changing over time and provide fold changes comparing the uninduced protein of interest with the IFN stimulated conditions.*

We thank the referee for suggestions to improve the clarity of figure 5. Accordingly we now show interactors that were filtered for proteins which were at least 2-fold enriched above background (mycBirA* or BirA*-NLS controls) in at least one condition at an adjusted p-value of <0.01, and which showed at least a 2-fold increase in intensity after interferon induction when compared to steady-state conditions. For this filtered set of proteins we computed the mean log₂ LFQ protein ratio of the interferon-induced (2h and 18h) and the steady-state condition and used these values to generate a hierarchical cluster analysis and heat-maps in Perseus with default settings.